# TIME-LAPSE: UNCERTAINTY SCORING VIA LATENT-SPACE EMBEDDINGS OVER TIME

## ABSTRACT

Safe deployment of trained ML models requires determining when they are uncertain of their predictions and refraining from using them for decision-making. Existing approaches inspect test samples in isolation to estimate their corresponding predictive uncertainty. However, in the real-world, deployed models typically see test inputs consecutively and predict labels continuously over time during inference. We propose TIME-LAPSE, an uncertainty scoring framework that examines a sequence of **la**tent-**sp**ace **e**mbeddings over **time** prior to the current sample to determine its predictive uncertainty. Specifically, (a) our spatial uncertainty score estimates uncertainty using distance metrics (Mahalanobis distance) and similarity metrics (cosine similarity) in the latent-space and (b) our temporal uncertainty score determines deviations in correlations over time using representations of past inputs in a non-parametric, sliding-window based algorithm. We evaluate TIME-LAPSE through the lens of out-of-distribution (OOD) detection and dataset shift detection on tasks over diverse domains: audio and vision using public datasets and further benchmark our approach on a challenging, real-world, electroencephalograms (EEG) dataset for seizure detection. We achieve state-of-the-art results for OOD detection in the audio and EEG domain and observe considerable gains in semantically corrected vision benchmarks. We show that TIME-LAPSE is more driven by semantic content compared to other methods, i.e., it is more robust to dataset statistics. We also show that TIME-LAPSE outperforms spatial methods significantly through our sequential evaluation framework that emulates real-life drift settings through extensive experiments and ablations.

## 1 INTRODUCTION

Modern machine learning (ML) has seen tremendous success in various tasks across multiple domains (Bojarski et al., 2016; Hinton, 2018; Kreinovich & Kosheleva, 2020; van den Oord et al., 2016), surpassing human performance in many benchmarks (Esteva et al., 2017; Yala et al., 2019; Krizhevsky et al., 2012). However, deep learning models have been known to fail silently and catastrophically with highly confident predictions (Nguyen et al., 2015; Goodfellow et al., 2015; Guo et al., 2017). Such models assume a closed world scenario, i.e., they assume that variability encountered when deployed in the real-world would be similar to the variability present in their training data. In practice, they encounter an open world where incoming samples can come from shifted distributions or completely new distributions (Liu et al., 2020). This behaviour can have severe consequences in mission-critical domains such as healthcare and autonomous driving, where errors can be costly, resulting in injury or even death (Amodei et al., 2016). Most widely cited models do not come with an uncertainty scoring mechanism to say "I don't know" or to abstain from prediction (Kompa et al., 2021). As deployed ML models are used to inform real-world decisions, it is important that they have the ability to understand when they ought to be "unsure". A good uncertainty scoring framework must assign higher uncertainty estimates when the network is faced with distributional shifts or new semantic content, i.e., out-of-distribution (OOD) data but must be certain (low uncertainty) and generalize well on learnt distributions yet unseen data (in-distribution or InD data).

The task of identifying when incoming inputs are drawn far from the training distribution is called out-of-distribution (OOD) detection. At its core, it is a binary classification problem, evaluated using measures such as area under the receiver operating curve (AUROC), area under the precision recall curve (AUPR) and false positive rate at 80% true positive rate (FPR80). Current OOD evaluation

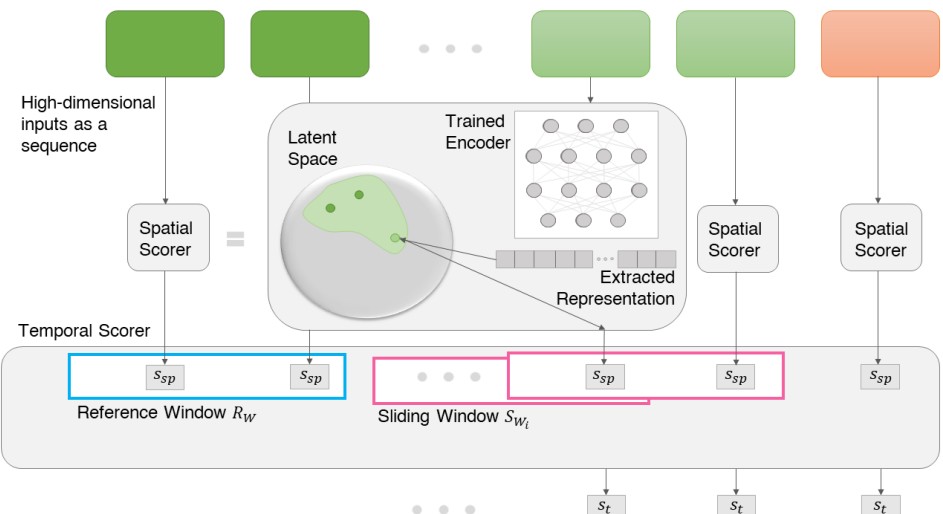

Figure 1: TIME-LAPSE for uncertainty scoring. High-dimensional inputs (images, audio, speech, EEG signals, etc) are passed in sequentially during inference. Dark green indicates samples from the training set. Light green and light orange represent potential InD and OOD inputs respectively. Each input is projected on to a lower-dimensional latent space using a well-trained encoder and its spatial uncertainty score $s_{sp}$ is extracted by the spatial scorer. The temporal scorer utilizes the sequence of spatial scores to give the final temporal uncertainty scores $s_t$.

schemes hold different datasets to be OOD sources, without considering if the distinctions are semantic in nature (Ahmed & Courville, 2020) leading to methods that are insensitive to semantic overlap and biased to dataset statistics. Closely related is the task of dataset shift detection (Moreno-Torres et al., 2012), crucial for deployed systems. Dataset shift can occur due to sample selection bias, non-stationary environments, missing values, new concepts evolving over time, etc. Most ML systems do not evaluate their ability to detect such shifts, presenting a huge gap between models performing well on test sets and models capable of being deployed in the wild.

In this paper, we propose TIME-LAPSE, a predictive uncertainty quantification framework using **la**tent **sp**ace **e**mbeddings over **time**, evaluated through downstream tasks of OOD detection and dataset shift detection. A key insight in our approach is that most real-world scenarios (such as electroencephalogram (EEG) seizure analysis, healthcare decision making, autonomous driving, etc.) involve consecutive inputs to the model that are likely to be correlated over time. For example, an obstacle detector deployed in a self-driving car will see images correlated over time. A seizure detector installed in a neurology clinic will process hours of time-correlated EEG signals. Even low-risk image classification models deployed in search engines over the cloud will see correlated inputs over time when grouped by user ID or location. Thus, sequentially occurring samples share meaningful semantic correlations.

In TIME-LAPSE (Fig. 1), we compute spatial uncertainty scores based on the following hypotheses: an encoder with enough inductive bias likely (a) maps OOD inputs "far" from other InD samples in the latent space under the Mahalanobis distance metric (Lee et al., 2018) and (b) produces dissimilarities between their representations under the cosine similarity metric (Jones & Furnas, 1987). We compute temporal uncertainty scores by exploiting correlations between lower dimensional representations of consecutive samples over time when examined as a sequence. A sequence of InD inputs will show more correlation than a sequence with both InD and OOD inputs or a sequence with just OOD inputs.

Recent works limit themselves to OOD detection tasks with evaluations purely on highly curated image benchmarks, but do not extend their tests to more diverse and realistic scenarios required for practical applications (Ren et al., 2019; Ahmed & Courville, 2020). Such OOD image benchmarks consider different datasets to be OOD sources, neglecting any semantic overlap between their OOD test sets and InD train sets, leading to narrow model capabilities (Yang et al., 2021). We show that TIME-LAPSE is more driven by semantic content compared to other techniques, i.e. it is more

robust to dataset statistics. Moreover, current evaluation schemes do not consider different dataset drift settings that could occur in the real-world. We put forth an additional sequential evaluation framework to evaluate methods under realistic conditions of data drifts and show that TIME-LAPSE outperforms other methods significantly. We demonstrate the efficacy of our proposed method by comparing against multiple baselines over different domains (vision, audio and clinical EEGs). We achieve state-of-the-art (SOTA) on both audio and EEG datasets while improving performance on semantically corrected image benchmarks. To the best of our knowledge, we are the first to use temporal sequences of latent space embeddings for uncertainty quantification. We are also the first to use deep-learning based uncertainty scoring techniques for EEG analyses.

To summarize, our key contributions are:

- We propose TIME-LAPSE, an uncertainty scoring framework that uses samples' similarity in the latent space and their temporal similarity amongst sequential inputs to determine their predictive uncertainty.

- We show that scores from TIME-LAPSE are driven by semantic content and are more robust to dataset statistics whereas popular baselines, benchmarked on standard image datasets, are susceptible to dataset statistics and overlook semantic overlap for OOD detection.

- We benchmark TIME-LAPSE on diverse domains: audio speech classification, seizure detection using clinical EEGs and image classification. TIME-LAPSE achieves state-of-the-art (SOTA) in audio tasks, the EEG domain and semantically corrected image benchmarks.

- We propose a sequential evaluation framework for dataset shift detection to evaluate methods under realistic conditions of data drifts and show that TIME-LAPSE outperforms other methods. We hope this evaluation scheme will be adopted for routinely characterizing ML system performance in the wild.

## 2 RELATED WORK

Uncertainty estimation is a rich field with a long history. Classical techniques include density estimation (Breunig et al., 2000), one-class SVMs (Schölkopf et al., 1999), tree-isolation forests (Liu et al., 2008), etc for one-dimensional data such as time-series However, such techniques scale badly with input dimensionality (Rabanser et al., 2019). Calibration is a frequentist notion of uncertainty, measured by proper scoring rules like log-loss or Brier scoring (Guo et al., 2017; DeGroot & Fienberg, 1983; Dawid, 1982). Uncertainty estimation for deep neural networks (NNs) (Lakshminarayanan et al., 2017) typically use a Bayesian formalism to learn distributions over model weights (Blundell et al., 2015; Malinin & Gales, 2018; Chen et al., 2019; Graves, 2011; Neal, 1996; Welling & Teh, 2011). However, most Bayesian methods are difficult to train and computationally expensive. Gal & Ghahramani (2016) use Monte-Carlo dropout at test-time with an approximate Bayesian interpretation to derive uncertainty estimates.It is also computationally expensive, requiring $k$ forward passes for each instance during inference. Quality of uncertainty estimates are commonly evaluated using OOD detection. Moreno-Torres et al. (2012) give a complete overview of closely related topics of distributional shift detection including covariate shift, label shift and concept drift (Gama et al., 2014). Several works consider semi-supervised techniques for OOD such as outlier exposure methods (Hendrycks et al., 2019a; Ruff et al., 2020), while many assume the supervised setting where the target OOD distribution is known and propose alternative training strategies and auxillary OOD tasks (DeVries & Taylor, 2018; Hendrycks et al., 2019b; Mohseni et al., 2020; Shalev et al., 2018) in a multi-task setting. In this paper, we consider the more general setting of unsupervised OOD detection where no OOD examples are labelled as such and the encoder has access to only InD training labels. Other methods use representations from NNs to infer OOD inputs. Hendrycks & Gimpel (2017) use maximum softmax probabilites (MSP) to detect OOD data. Liang et al. (2018) improve this by introducing a temperature parameter to the softmax equation. Lee et al. (2018) fit class-conditional Gaussians to intermediate activations and use the Mahalanobis distance to identify OOD samples. Recently, some works use self-supervision to get better representations to improve OOD detection (Tack et al., 2020; Winkens et al., 2020; Sehwag et al., 2021). Several methods directly use input likelihoods as the OOD detection score. However, studies have shown that generative techniques may be overconfident on complex inputs (Nalisnick et al., 2019). Modified likelihood scores have been proposed since then (Serrà et al., 2020; Xiao et al., 2020; Choi et al., 2019) including energy-based models (Liu et al., 2020; Du & Mordatch, 2019; Grathwohl et al., 2020) and likelihood ratios (Ren

et al., 2019). Reconstruction-based OOD methods use reconstruction loss as the uncertainty score and primarily use auto-encoders (Zong et al., 2018; Pidhorskyi et al., 2018) or GANs (Schlegl et al., 2017; Deecke et al., 2019; Perera et al., 2019). Our temporal uncertainty scores builds upon and shares similarity to change point detection methods in time-series data (Aminikhanghahi & Cook, 2017; Kifer et al., 2004) along with signal processing techniques such as Particle and Kalman filtering (van der Merwe et al., 2001; Kalman, 1960).

## 3 TIME-LAPSE: OUR UNCERTAINTY SCORING FRAMEWORK

We propose TIME-LAPSE, an uncertainty scoring framework using **la**tent-**sp**ace **e**mbeddings over **time**. We consider a multi-class classification setting here, though our framework can be extended to other scenarios such as regression, segmentation, etc.

### 3.1 PROBLEM SETUP

Let $\mathcal{X}$ represent our high-dimensional input space, $\mathcal{X} \subseteq \mathbb{R}^n$. Let $\mathcal{Y} = \{0, 1, 2, ..., C-1\}$ denote the label space where $C$ is the number of classes. In standard multi-class classification, we learn a classifier $f : \mathcal{X} \mapsto \mathcal{Y}$ using a dataset $D_{train}$ (assumed to be sampled from an underlying distribution $p^*$) such that $f(x) = p(Y = y_i|x)$, where $x \in \mathcal{X}$ and $y_i \in \mathcal{Y} \; \forall \; i$. The final prediction for an unseen input $x$ is given by $\hat{y} = \arg\max_{y_i} p(y_i|x)$. Given a sequence of unseen inputs, $x_t$ (where $t$ denotes time) to a classifier during inference, our goal is to output a score $s(x_t)$ and a selective function $g(x_t)$ with threshold th $\forall \; x_t$ such that

$$(g, s)(x_t) := \begin{cases} \hat{y} = \arg\max_{y_i} p(y_i|x_t), & \text{if } s(x_t) \leq \text{th} \\ \text{ABSTAIN or FLAG AS OOD}, & \text{else} \end{cases}$$

We call the score $s(x_t)$ the associated uncertainty score of input sample $x_t$

### 3.2 TIME-LAPSE FRAMEWORK

We use the learnt classifier $f$ to derive an encoder $h : \mathcal{X} \mapsto \mathcal{U}$ that maps high-dimensional inputs $x \in \mathcal{X} \subseteq \mathbb{R}^n$ onto a latent space $\mathcal{U} \subseteq \mathbb{R}^d$ through its intermediate representations. Note that $h$ is inherently learnt when $f$ is trained using the InD training dataset $D_{train}$. TIME-LAPSE is agnostic to the training method, which can be supervised, self-supervised or unsupervised depending on how the label information in $D_{train}$ is utilized.

#### 3.2.1 LEARNING SPATIAL UNCERTAINTY SCORES

To extract the spatial uncertainty score of an input test sample's prediction, we apply two fundamental hypotheses: (a) a well-trained encoder maps InD samples onto a dense region in the latent space but maps OOD inputs outside this region under a distance metric $d$ with high likelihood, and (b) the OOD inputs that get mapped to the latent space do not share similarity with InD inputs under a similarity metric, $sim$.

To identify spatially distinguishable OOD inputs, we first model the extent of InD region $\mathfrak{D}$ within the fixed-dimensional latent space, using the InD samples. We extract a coreset by sampling from our InD training data, $\mathcal{D}_{train}$.

$$\text{coreset} = \{h(x_i) \mid x_i \sim \mathcal{D}_{train}\}; \quad |\mathcal{D}_{train}| \geq |\text{coreset}|$$

We compute the distance score $s_{\text{dist}}$ and the similarity score $s_{\text{sim}}$ of the unseen test sample $x$ by comparing its latent representation to that of samples in the coreset using a distance metric $d$ and a similarity metric $sim$ such that:

$$s_{\text{dist}}(x) = \min_{h(x_i) \, \in \, \text{coreset}} d(h(x_i), h(x)) \; ; \; s_{\text{sim}}(x) = \max_{h(x_i) \, \in \, \text{coreset}} sim(h(x_i), h(x))$$

Finally, the combined spatial uncertainty score is given by

$$s_{\text{spatial}}(x) = s_{\text{dist}}(x) \, . \, s_{\text{sim}}(x)$$

We choose the distance metric $d$ to be the Mahalanobis distance obtained using class-conditional Gaussians (Lee et al., 2018) without label smoothing (Winkens et al., 2020) where class-wise means $\mu_c$ and covariance matries $\Sigma_c$ are estimated from the coreset.

$$d(h(x)) = \min_c \ (h(x) - \mu_c)^T \, \Sigma_c^{-1} \, (h(x) - \mu_c)$$

As similarity metric $sim$, we adopt the cosine similarity between the test input's representation and the coreset.

$$\text{sim}(h(x), h(x_i)) = \frac{h(x) \cdot h(x_i)}{||h(x)|| \ ||h(x_i)||}; \quad h(x_i) \in \text{coreset}$$

We note that TIME-LAPSE will accept any $d$ and $sim$ that is most suited for the target domain.

### 3.2.2 LEARNING TEMPORAL UNCERTAINTY SCORES

Our hypothesis in extracting our temporal uncertainty scores is that there is value in evaluating samples in the context of other samples instead of viewing them independently. In our framework, we consider high-dimensional inputs to the model as a sequence (Fig. 1). We use the representational capabilities of modern encoders and examine them over time instead of directly modeling the incoming high-dimensional, multivariate data stream or treating each sample independently, i.e., we detect when input samples go out-of-distribution by comparing their reduced-dimension scores ($s_{\text{spatial}}$) with those of other inputs over time. We note that when these scores are scalar (as in our case), the problem reduces to change-point detection for a one-dimensional sequence.

We put forth an unsupervised, non-parametric, sliding-window based algorithm that builds on the work done by Kifer et al. (2004) to generate our temporal uncertainty scores $s_{\text{temporal}}$. We consider a sequence of $r$ samples in the input data stream denoted by $\{x_1, x_2, ..., x_t, ..., x_r\}$. We obtain the scalar spatial scores for the $r$ samples, $\{s_{\text{spatial}}(x_1), s_{\text{spatial}}(x_2), ..., s_{\text{spatial}}(x_t), ..., s_{\text{spatial}}(x_r)\}$ from their representations $\{h(x_1), h(x_2), ..., h(x_t), ..., h(x_r)\}$ as explained in Section 3.2.1.

Given two window sizes $w_A$ and $w_B$ ($w_A, w_B < r$), we define a reference window $R_W$ and a sliding window $(S_W)_t$ to be

$$R_W = [s_{\text{spatial}}(x_1), ..., s_{\text{spatial}}(x_{w_A})]$$
$$(S_W)_t = [s_{\text{spatial}}(x_{(w_A + t))}), ..., s_{\text{spatial}}(x_{(w_A + t + w_B)})]$$

where $t \in \{1, ..., (r - w_A - w_B)\}$ as visualized in Fig. 1. On choosing an appropriate statistic or distance measure $\mathcal{F}$ (e.g. probability odds ratio, Kolmogorov-Smirnov, Kullback-Leibler, Wilcoxon, Mann-Whitney, etc.), we compute the temporal uncertainty score between reference window $R_W$ and the $t^{th}$ sliding window $(S_W)_t$ for the $t^{th}$ sample

$$s_{\text{temporal}}(x_t) = \mathcal{F}(R_W, (S_W)_t)$$

The null hypothesis $\mathcal{H}_0$ for our setting here is that the two windows (reference and sliding) are similar and come from the same distribution. We hope to reject $\mathcal{H}_0$ in favour of the alternate hypothesis $\mathcal{H}_1$ at a significance level of 0.05 whenever the inputs go out-of-distribution. This approach (as in (Kifer et al., 2004)) assumes the data points are generated sequentially by some underlying probability distribution, but otherwise makes no assumptions on the nature of the generating distribution nor does it assume the samples are identically distributed.

## 4 EXPERIMENTAL SETUP

We perform experiments across multiple domains (vision, audio and clinical EEG signals) to evaluate our approach. We choose the task of image classification for the visual domain, spoken word classification from audio clips for the audio domain and the challenging, real-world clinical task of seizure detection from EEG signals framed as an EEG clip binary classification task for the healthcare domain. Our code will be made publicly available after publication.

We use a variety of datasets and models in our experiments. For the audio domain, we use the Free Spoken Digits Dataset (FSDD) (Jackson et al., 2017) and Google Speech Commands (GSC) dataset (Warden, 2018). We train an M5 encoder (Dai et al., 2016) with raw audio InD data from

spoken digits 0-9 from GSC and evaluate it on unseen InD samples from GSC 0-9 and FSDD 0-9 along with OOD samples from the non-digit spoken words from GSC. For our seizure tasks, we use EEG data from Stanford Health Care (SHC) (InD: patient population 20-60 years), Lucile Packard Children's Hospital (LPCH) (OOD: patient population <20 years) and the public Temple University Hospital seizure corpus (TUH) (Shah et al., 2018; Obeid & Picone, 2016) (OOD: different patient demographics) along with a Dense-Inception encoder as described by Saab et al. (2020). For our vision tasks, we use MNIST as InD and MNIST-like data as OOD and CIFAR10 as InD, SVHN as OOD and CIFAR100 as a semantic OOD evaluation set up. Details on the data, model training, embedding extraction and temporal framework settings in Appendix B.1 and B.3.

We compare against six baselines: Maximum Softmax Probability (MSP) (Hendrycks & Gimpel, 2017), Predictive Entropy (Ren et al., 2019), KL Divergence with Uniform Distribution (Hendrycks et al., 2019a), ODIN (Liang et al., 2018), Vanilla Mahalanobis distance (Lee et al., 2018) and Test-time Dropout (Gal & Ghahramani, 2016). We provide baseline details in Appendix B.2

### 4.1 Evaluation

For all experiments, we evaluate our approach only using samples unseen during training. Our test sets include unseen samples drawn from the same distribution as the training data along with samples from unseen classes and significantly different datasets. We also follow the setting recommended by Ahmed & Courville (2020) to avoid dataset bias by holding out few classes from a dataset during training and using the held-out classes as new semantic content for evaluation. We evaluate OOD detection performance using standard metrics: area under the receiver operating characteristic (AUROC↑) , area under the precision-recall curve (AUPR↑) and the false positive rate at $80\%$ true positive rate (FPR80↓) as commonly prescribed (Ren et al., 2019).

It is important to note that what constitutes InD or OOD is driven by context, i.e., it depends on the target task, the encoder capacity, what models need to be robust to and what they should detect as outliers. Standard OOD benchmarks treat different datasets to be OOD resulting in models with narrow capabilities susceptible to dataset bias (Ahmed & Courville, 2020; Yang et al., 2021). Moreover, they fail to capture any semantic overlap between different datasets, e.g., if a model is trained to classify cats from dogs and is shown a breed of dog (high semantic overlap) from a different dataset during inference, it should be considered InD whereas a muffin would be OOD since it has no semantic overlap with the train set. We evaluate OOD detection performance with & without semantic overlap and show that TIME-LAPSE is driven more by semantic content than other baselines.

We further propose a sequential evaluation framework to evaluate uncertainty estimation under realistic conditions of data drifts. We evaluate performance metrics such as $\%$ error across trials and detection accuracy over $N = 1000$ trials. For each trial, we generate a stream of test data from our test sets of length 10,000 samples with change points inserted every $k$ samples in each trial. Distribution changes are simulated by first randomly choosing InD or OOD (Bernoulli random variable with probability $p \in \{0.2, 0.5, 0.7\}$) and then randomly drawing $k \in \{50, 100, 200, 500, 1000, 5000\}$ samples from the chosen distribution. We emulate various drift conditions including rapidly changing streams and slowly shifting streams.

## 5 Results

### 5.1 OOD Detection Performance on Audio, Clinical EEG and Image datasets

We achieve the state-of-the-art (SOTA) with TIME-LAPSE scores on OOD detection in our audio and EEG-based tasks of spoken digits classification and seizure detection compared to our baselines (Table 1) across all metrics: AUROC, AUPR and FPR80. The AUROC values for TIME-LAPSE scores are $73.9\%$ and $77.1\%$, 9 points and 12 points higher than the strongest baseline, Test-time Dropout, for both tasks respectively. In our MNIST experiments, we see TIME-LAPSE performs comparably with Test-Time Dropout. It has the best AUPR and comparable AUROC and FPR80 values. Over CIFAR10 experiments with the SVHN dataset (no semantic overlap with CIFAR10) as OOD, TIME-LAPSE outperforms other baselines significantly. When CIFAR100 classes are added (considered OOD as in standard benchmarks) to the evaluation, TIME-LAPSE shows the best AUPR score and the second best AUROC and FPR80. We examine the effect of semantic overlap present in CIFAR100 classes with the InD classes as well as those of other domains in Section 5.2.

Table 1: OOD performance. Mean scores AUROC ↑ / AUPR ↑ / FPR80 ↓ over 5 random runs are reported. Standard Deviations are reported in Appendix (Tables 5, 6, 7). Best scores in bold.

| | Audio | EEG Data | Vision | | |
|---|---|---|---|---|---|
| Task | Speech Classification | Seizure Detection | Image Classification / Digit Classification | | |
| OOD Sets | Other spoken words | Other institutions | SVHN | CIFAR100 | x-MNIST |
| MSP | 0.626 / 0.527 / 0.515 | 0.358 / 0.421 / 0.754 | 0.760 / 0.770 / 0.358 | 0.852 / 0.986 / 0.231 | 0.899 / 0.979 / 0.148 |
| Predictive Entropy | 0.615 / 0.515 / 0.515 | 0.393 / 0.495 / 0.742 | 0.761 / 0.752 / 0.357 | 0.854 / 0.986 / 0.230 | 0.902 / 0.981 / 0.147 |
| KL_U | 0.553 / 0.475 / 0.579 | 0.390 / 0.472 / 0.719 | 0.775 / 0.786 / 0.347 | 0.860 / 0.987 / 0.223 | 0.899 / 0.980 / 0.164 |
| ODIN | 0.466 / 0.448 / 0.712 | 0.325 / 0.388 / 0.790 | 0.748 / 0.776 / 0.402 | 0.845 / 0.986 / 0.251 | 0.898 / 0.979 / 0.148 |
| Vanilla Mahalanobis | 0.680 / 0.636 / 0.520 | 0.633 / 0.651 / 0.525 | 0.738 / 0.782 / 0.477 | 0.679 / 0.967 / 0.558 | 0.918 / 0.984 / 0.118 |
| Test-Time Dropout | 0.649 / 0.619 / 0.523 | 0.647 / 0.619 / 0.583 | 0.716 / 0.725 / 0.494 | **0.925** / 0.986 / **0.049** | **0.976** / 0.986 / **0.016** |
| TIME-LAPSE (ours) | **0.739 / 0.704 / 0.439** | **0.771 / 0.701 / 0.335** | **0.814 / 0.827 / 0.311** | 0.866 / **0.988** / 0.239 | 0.972 / **0.995** / 0.037 |

Table 2: Effect of semantic content on OOD performance. In spoken digit classification, spoken digits from different datasets (high semantic overlap) as InD should result in AUROC ↑. Naively considering them to be OOD (dataset driven) should result in AUROC ↓ Mean and standard deviation over 5 random runs. Best in bold.

| | MSP | Predictive Entropy | KL_U | ODIN | Vanilla Mahalanobis | Test-Time Dropout | TIME-LAPSE (ours) |
|---|---|---|---|---|---|---|---|
| Semantic OOD | $0.621 \pm 0.006$ | $0.611 \pm 0.006$ | $0.550 \pm 0.006$ | $0.466 \pm 0.007$ | $0.676 \pm 0.016$ | $0.644 \pm 0.005$ | $\mathbf{0.739 \pm 0.006}$ |
| Dataset Driven | $0.818 \pm 0.005$ | $0.821 \pm 0.005$ | $0.806 \pm 0.006$ | $0.675 \pm 0.006$ | $\mathbf{0.495 \pm 0.008}$ | $0.821 \pm 0.002$ | $\mathbf{0.506 \pm 0.007}$ |

## 5.2 AFFINITY FOR SEMANTIC CONTENT AND ROBUSTNESS TO DATASET STATISTICS

We study the effect of semantic overlap in OOD classes and characterize how TIME-LAPSE compares against other methods in its robustness to dataset statistics over multiple domains.

**Audio** In our audio experiments, we train our encoder only on spoken digits 0-9 (forming the InD classes) from the Google Speech Commands (GSC) dataset. The other 25 classes (Yes, No, Right, Left, Bird, etc) in GSC have never been encountered by the model and have drastic semantic shifts in their content when compared to the InD classes 0-9 and are considered OOD. During evaluation, we show the model unseen samples from each of the classes in GSC along with samples from the Free Spoken Digits Dataset (FSDD) which contains classes 0-9 (same semantic content as InD data but different dataset) and measure OOD detection performance (Table 2). As an ablation, we examine the performance when we flip the FSDD classes to be OOD data in our evaluation. We see that AUROC, AUPR and FPR80 values for most baselines (notably Test-time Dropout and Predictive Entropy) increase significantly, whereas they drop for TIME-LAPSE and vanilla Mahalanobis (Table 2). This shows that our strongest baseline methods have narrow capabilities that assume only data belonging to InD classes *and* having the *same dataset statistics* as the training data will be considered as InD. All other data will be rejected as OOD, leading to very high FPRs in practice. In contrast, TIME-LAPSE is able to identify true InD samples despite differing dataset statistics, i.e. TIME-LAPSE detects OOD samples based on semantic content than dataset statistics when compared to the other methods.

**Vision (CIFAR10)** For our CIFAR10 experiments, we train our encoder only on 7 classes of CIFAR10, leaving out classes Airplane, Bird and Dog. SVHN data consists of images of numbers and has no semantic overlap with any of the 7 InD classes (Automobile, Cat, Deer, Frog, Horse, Ship, Truck). To study the effect of semantic overlap, we consider the 100 classes from CIFAR100 along with the 3 left out classes from CIFAR10. We compare the performance metrics when naively considering all of the above as OOD ignoring any semantic overlap. We see that baselines perform better while TIME-LAPSE drops (Table 1). We analyse class-wise performance and we see that TIME-LAPSE's performance strongly correlates to semantic content compared to other methods. For example, consider Class Pickup-Truck and Streetcar from CIFAR100. While they are disjoint from the InD classes Automobile and Truck in CIFAR10, they shares a high degree of semantic content and should be considered InD. When models encounter samples from classes that have high semantic overlap with InD, they need to consider them InD and not OOD. Instead, if such samples are considered OOD, only methods that have narrow model capacities will show high detection performance (Table 3) indicating their dependence on data statistics instead of semantic content.

**Vision (MNIST)** We train two different encoders, one trained on all digits 0-9 while the other was trained on $\{0, 1, 4, 6, 7, 8, 9\}$ leaving digits 2, 3 and 5. Amongst our evaluations, only certain classes

from the e-MNIST data shared semantic overlap – letters 'o','l','i','z','y','s' and 'q' share structural similarity with classes 0, 1, 2, 4, 5 and 9 respectively. We show results for both encoders with and without the above classes in the OOD evaluations (Table 4).

**Clinical EEGs** While discussions on semantic issues on clinical EEGs may be out of scope for this paper, we do observe interesting semantic effects in this case as well. For instance, seizure types unseen by the network will generate higher uncertainty scores.

Table 3: Pickup Truck & Street Car share high semantic overlap with InD classes Truck & Automobile. Evaluations when naively considering them to be OOD should result in AUROC ↓ / AUPR ↓. Airplane & Fish do not show any semantic overlap with InD classes, should simultaneously result in AUROC ↑ / AUPR ↑

| CIFAR | MSP | Predictive Entropy | KL_U | ODIN | Vanilla Mahalanobis | Test-Time Dropout | TIME-LAPSE (ours) |
|---|---|---|---|---|---|---|---|
| Pickup Truck | 0.794 / 0.292 | 0.791 / 0.276 | 0.698 / 0.181 | 0.815 / 0.182 | 0.691 / 0.207 | 0.942 / 0.506 | **0.688 / 0.183** |
| Street Car | 0.767 / 0.255 | 0.768 / 0.256 | 0.766 / 0.239 | 0.785 / 0.255 | 0.656 / 0.181 | 0.838 / 0.337 | **0.757 / 0.256** |
| Airplane | 0.891 / 0.488 | 0.894 / 0.519 | 0.913 / 0.571 | 0.893 / 0.549 | 0.658 / 0.192 | 0.955 / 0.547 | **0.902 / 0.575** |
| Aquarium Fish | 0.893 / 0.492 | 0.897 / 0.529 | 0.913 / 0.589 | 0.888 / 0.526 | 0.705 / 0.209 | 0.955 / 0.542 | **0.911 / 0.577** |

Table 4: MNIST models. 0to9 denotes encoder trained on 0-9 digits. M235 indicates encoder trained on digits {0 to 9} - {2,3,5}, i.e., {0,1,4,6,7,8,9}. Semantic means that e-MNIST classes with ambiguous semantic overlap on InD classes, i.e., {l,i,o,..} are removed from OOD evaluation

| MNIST | MSP | Predictive Entropy | KL_U | ODIN | Vanilla Mahalanobis | Test-Time Dropout | TIME-LAPSE (ours) |
|---|---|---|---|---|---|---|---|
| 0to9 | 0.895 / 0.968 | 0.899 / 0.971 | 0.903 / 0.972 | 0.896 / 0.969 | 0.903 / 0.973 | 0.963 / 0.973 | 0.961 / 0.990 |
| 0to9 Semantic | 0.921 / 0.970 | 0.926 / 0.973 | 0.928 / 0.975 | 0.922 / 0.971 | 0.937 / 0.976 | 0.976 / 0.970 | **0.986 / 0.995** |
| M235 | 0.907 / 0.982 | 0.910 / 0.983 | 0.909 / 0.983 | 0.906 / 0.982 | 0.919 / 0.985 | 0.976 / 0.986 | 0.975 / 0.996 |
| M235 Semantic | 0.926 / 0.983 | 0.929 / 0.984 | 0.925 / 0.983 | 0.925 / 0.983 | 0.941 / 0.986 | 0.984 / 0.986 | **0.990 / 0.998** |

## 5.3 DRIFT DETECTION: SEQUENTIAL EVALUATION RESULTS

To evaluate the performance under our sequential framework, we define the detector to have erred if it wrongly concludes an incoming test sample $x_t$ to be OOD when it is actually InD or vice-versa. If the framework detects the change point within its fixed window size, we do not consider it as an error. We generate data streams of length 10,000 samples changing the distribution every $k$ samples in each trial The distribution changes are simulated by first randomly choosing InD or OOD (Bernoulli random variable with probability $p \in \{0.2, 0.5, 0.7\}$) and then randomly drawing the $k \in \{50, 100, 200, 500, 1000, 5000\}$ samples from the chosen distribution. We calculate the TIME-LAPSE scores using the two-sided Mann-Whitney test (Mann & Whitney, 1947) at a significance level of 0.05 for each sample in the stream and run it through our sequential detector with window sizes $w_A = w_B = 25$ for vision experiments and $w_A = w_B = 50$ for audio and EEG experiments. We show example plots of datastreams for each domain in the Appendix, Figs. 2, 3, 4, 5, 6, 7, 8, 9. We perform a 2-cluster KMeans on the generated Mann-Whitney scores to assign InD and OOD predictions. We then calculate the cumulative error by adding the errors of all samples to calculate detection error. We repeat this experiment $N$ times ($N = 1000$ here) to estimate the detector's error distribution (EEG distribution given in the Fig. 2, Appendix). Over 73% of the cumulative errors in our EEG task (over 85% for our audio task) lie within 10% and over 93% of the errors in our EEG task (over 97% for our audio task) lie within 20% indicating our detector's high performance in comparison to other spatial detectors. For our vision tasks, the error distribution is much tighter and we are able to get near 99% detection accuracy for over 95% of the time.

## 5.4 ABLATIONS: EFFECT OF ENCODER CAPACITY

We perform extensive ablations on TIME-LAPSE to study (i) the effect of individual scorers: distance, similarity, spatial, temporal on performance (Appendix D), (ii) the effect of the coreset size on performance (Appendix E), (iii) the effects of encoder capacity on performance (here, Section 5.4, Appendix F), and (iv) qualitative analyses (Appendix G).

To study the effects of encoder capacity on TIME-LAPSE, we consider our audio task. We compare two encoders, trained on the same InD data and evaluated with the same OOD data. The lower capacity encoder, Kymatio, is a two-layer network with a log-scattering layer that extracts scattering coefficients from the inputs as latent space embeddings, followed by a log-softmax layer for classification with 0.75 classification accuracy for InD test samples. The higher capacity encoder is a CNN-based M5 model with 0.85 classification accuracy for InD test samples. As shown in Fig. 10 (Appendix F), Kymatio is not able to separate out any of the classes based on TIME-LAPSE scores whereas M5 is able to correctly separate out semantic InD samples from OOD samples well. Thus, the success of TIME-LAPSE is highly dependent on the capacity of the model.

To further investigate the effects of the lower capacity model, we train Kymatio for the same task (i.e., same InD semantic classes 0-9) but with the much smaller FSDD dataset. The model reaches 0.97 classification accuracy, showing that it is able to express high-dimensional audio signals well. However, on comparing its TIME-LAPSE scores with those of Kymatio trained on GSC 0-9 (Fig 11, Appendix F), we see that the former separates out FSDD InD from GSC InD and GSC OOD, indicating that it is overfitting to dataset statistics whereas the latter is not able to separate out any of them. Thus, TIME-LAPSE scores can help in determining both OOD detection capability of an encoder as well as generalization capability over OOD samples.

## 6 DISCUSSION

In this work, we develop a unique uncertainty scoring framework that can identify when incoming inputs fall out-of-distribution with respect to trained encoders and subsequently enable models to abstain from prediction on such inputs. Our theoretical formulation encompasses all representation-based OOD detection methods and provides an opportunity to leverage temporal signals in combination. We show that longitudinal information across time can be leveraged for the challenging task of identifying distributional shifts. Through our ablations (Appendix D), we verify that temporal uncertainty scores and spatial uncertainty scores are complementary and combining them gives additional information than when used separately. We use latent-space embeddings to determine spatial uncertainty and proposed a simple sliding-window hypothesis-testing based approach to model temporal uncertainty But TIME-LAPSE can accept any type of spatial scoring and temporal scoring. We believe that there is scope for developing sophisticated methods that leverage the temporal aspect and leave it open for future work.

Furthermore, alongside works by Ahmed & Courville (2020) and Yang et al. (2021), we motivate the necessity of rethinking standard OOD image benchmarks that disregard semantic overlaps. We note that the task of identifying OOD inputs is highly dependent on context. For a classifier trained to identify cats and dogs, a tree is OOD whereas all of them would be InD for an object recognition system in a self-driving car. We believe that showing generalizability of OOD detection methods across domains is key to ensure robust performance in the wild. It is not sufficient to show better performance over just image OOD benchmarks. We show that TIME-LAPSE generalizes across 3 diverse domains- vision, audio and healthcare EEG, through our extensive experiments and analyses. We also propose a sequential evaluation framework to view OOD detection from the lens of dataset drift detection. To enable safe deployment and periodic monitoring of trained models, we believe that such a sequence-based evaluation scheme is essential. We hope this evaluation scheme will be adopted for routinely characterizing uncertainty estimation and model performance in the wild.

For our experiments, we train encoders with labelled InD data in a supervised setting. But our framework will support and benefit from more robust representations (Section 5.4, Appendix F) learnt via self-supervision or semi-supervision. We also note that both the audio and seizure analyses generate much higher-dimensional inputs compared to image classification. We hypothesize that model performance is strongly dependent on semantic content for such high-dimensional inputs when compared to dataset statistics and thus, gains from TIME-LAPSE are more obvious.

In conclusion, we provide a framework that utilizes both latent-space and time to determine predictive uncertainty for deep learning models. Our method is end-to-end trainable and network agnostic: it (a) does not require exposure to outliers (though can easily be extended to accommodate them if present), (b) does not require modifications to existing network architectures, and (c) does not require computationally heavy dropout techniques or ensemble passes. It can thus can be easily adopted to high-risk settings such as clinical workflows and autonomous driving.

ETHICS STATEMENT

We strongly believe that uncertainty quantification and its associated tasks of out-of-distribution detection and dataset need to be benchmarked and evaluated on diverse datasets and domains aside from standard, well-curated, image datasets to reduce the effect of bias, variability, "easyness" of the task or domain. To that effect, we have chosen three disparate domains (audio, vision and clinical EEGs) and associated tasks (spoken word classification, image classification, seizure detection posed as binary classification). While this is an important and necessary step, we believe our work and the community at large will only benefit from evaluating on more domains, datasets and tasks to ascertain a method's success and failure modes to ensure model deployment. We also hope to work with domain experts to validate TIME-LAPSE's strengths and error modes for more real-world use cases. All the data used in our work are either publicly available or are used with full IRB approval.

REPRODUCIBILITY STATEMENT

Our source code for TIME-LAPSE as well as our baseline implementations will be released to the public via Github upon publication. The datasets used for our audio and vision tasks are publicly available: FSDD (Jackson et al., 2017), GSC (Warden, 2018), $x$-MNIST (Lecun et al., 1998; Cohen et al., 2017; Tarin et al., 2018; Xiao et al., 2017), CIFAR10 & CIFAR100 (Krizhevsky & Hinton, 2009). The Temple University Hospital EEG Seizure Corputs (TUH) used in our clinical EEG experiments is publicly available (Shah et al., 2018; Obeid & Picone, 2016). EEG data from Stanford Healthcare (SHC) and Lucile Packard Children's Hospital (LPCH) contain patient sensitive information and cannot be shared publicly. Details on the data preprocessing steps and experimental settings are provided in Appendix B.

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

APPENDIX

## A SUPPLEMENTARY TABLES

Table 5: OOD performance: speech classification task (audio domain) and seizure detection (clinical EEGs) task. Mean and standard deviations over five random runs. Best scores indicated in bold.

| | Audio | | | Clinical EEGs | | |
|---|---|---|---|---|---|---|
| | AUROC ↑ | AUPR ↑ | FPR80 ↓ | AUROC ↑ | AUPR ↑ | FPR80 ↓ |
| MSP | $0.626 \pm 0.006$ | $0.527 \pm 0.005$ | $0.515 \pm 0.006$ | $0.358 \pm 0.008$ | $0.421 \pm 0.002$ | $0.754 \pm 0.022$ |
| Predictive Entropy | $0.615 \pm 0.006$ | $0.515 \pm 0.005$ | $0.515 \pm 0.009$ | $0.393 \pm 0.006$ | $0.495 \pm 0.012$ | $0.742 \pm 0.018$ |
| KL_U | $0.553 \pm 0.005$ | $0.475 \pm 0.002$ | $0.579 \pm 0.007$ | $0.390 \pm 0.003$ | $0.472 \pm 0.005$ | $0.719 \pm 0.010$ |
| ODIN | $0.466 \pm 0.006$ | $0.448 \pm 0.011$ | $0.712 \pm 0.092$ | $0.325 \pm 0.015$ | $0.388 \pm 0.022$ | $0.790 \pm 0.035$ |
| Vanilla Mahalanobis | $0.680 \pm 0.014$ | $0.636 \pm 0.009$ | $0.520 \pm 0.017$ | $0.633 \pm 0.028$ | $0.651 \pm 0.011$ | $0.525 \pm 0.025$ |
| Test-Time Dropout | $0.649 \pm 0.003$ | $0.619 \pm 0.007$ | $0.523 \pm 0.003$ | $0.647 \pm 0.004$ | $0.619 \pm 0.002$ | $0.583 \pm 0.012$ |
| TIME-LAPSE (ours) | $\mathbf{0.739 \pm 0.006}$ | $\mathbf{0.704 \pm 0.008}$ | $\mathbf{0.439 \pm 0.005}$ | $\mathbf{0.771 \pm 0.009}$ | $\mathbf{0.701 \pm 0.004}$ | $\mathbf{0.335 \pm 0.039}$ |

Table 6: OOD performance: CIFAR10 image classification on SVHN (no semantic overlap) vs CIFAR100. Mean and standard deviations over five random runs. Best scores indicated in bold.

| | SVHN | | | CIFAR100 | | |
|---|---|---|---|---|---|---|
| | AUROC ↑ | AUPR ↑ | FPR80 ↓ | AUROC ↑ | AUPR ↑ | FPR80 ↓ |
| MSP | $0.760 \pm 0.047$ | $0.770 \pm 0.021$ | $0.358 \pm 0.045$ | $0.852 \pm 0.014$ | $0.986 \pm 0.001$ | $0.231 \pm 0.018$ |
| Predictive Entropy | $0.761 \pm 0.047$ | $0.752 \pm 0.063$ | $0.357 \pm 0.045$ | $0.854 \pm 0.014$ | $0.986 \pm 0.001$ | $0.230 \pm 0.017$ |
| KL_U | $0.775 \pm 0.052$ | $0.786 \pm 0.029$ | $0.347 \pm 0.060$ | $0.860 \pm 0.015$ | $0.987 \pm 0.001$ | $0.223 \pm 0.023$ |
| ODIN | $0.748 \pm 0.068$ | $0.776 \pm 0.031$ | $0.402 \pm 0.095$ | $0.845 \pm 0.018$ | $0.986 \pm 0.001$ | $0.251 \pm 0.031$ |
| Vanilla Mahalanobis | $0.738 \pm 0.039$ | $0.782 \pm 0.029$ | $0.477 \pm 0.075$ | $0.679 \pm 0.012$ | $0.967 \pm 0.001$ | $0.558 \pm 0.025$ |
| Test-Time Dropout | $0.716 \pm 0.057$ | $0.725 \pm 0.042$ | $0.494 \pm 0.069$ | $\mathbf{0.925 \pm 0.004}$ | $0.986 \pm 0.000$ | $\mathbf{0.049 \pm 0.003}$ |
| TIME-LAPSE (ours) | $\mathbf{0.814 \pm 0.019}$ | $\mathbf{0.827 \pm 0.024}$ | $\mathbf{0.311 \pm 0.034}$ | $0.866 \pm 0.008$ | $\mathbf{0.988 \pm 0.001}$ | $0.239 \pm 0.028$ |

Table 7: OOD performance: Image classification on $x$-MNIST. Mean and standard deviations over five random runs. Best scores indicated in bold.

| | $x$-MNIST | | |
|---|---|---|---|
| | AUROC ↑ | AUPR ↑ | FPR80 ↓ |
| MSP | $0.899 \pm 0.006$ | $0.979 \pm 0.002$ | $0.148 \pm 0.012$ |
| Predictive Entropy | $0.902 \pm 0.006$ | $0.981 \pm 0.002$ | $0.147 \pm 0.011$ |
| KL_U | $0.899 \pm 0.007$ | $0.980 \pm 0.002$ | $0.164 \pm 0.012$ |
| ODIN | $0.898 \pm 0.006$ | $0.979 \pm 0.002$ | $0.148 \pm 0.015$ |
| Vanilla Mahalanobis | $0.918 \pm 0.006$ | $0.984 \pm 0.001$ | $0.118 \pm 0.015$ |
| Test-Time Dropout | $\mathbf{0.976 \pm 0.002}$ | $0.986 \pm 0.001$ | $\mathbf{0.016 \pm 0.001}$ |
| TIME-LAPSE (ours) | $\mathbf{0.972 \pm 0.002}$ | $\mathbf{0.995 \pm 0.000}$ | $0.037 \pm 0.038$ |

## B DETAILED EXPERIMENTAL SETUP

### B.1 DATASETS

**Audio** For the task of spoken word classification of audio recordings, we use the Free Spoken Digits Dataset (FSDD) (Jackson et al., 2017), consisting of .wav recordings of English digits 0-9, and the Google Speech Commands (GSC) dataset (Warden, 2018) that contain .wav recordings of 35 English keywords including the digits 0-9 and non-digits such as "Yes", "No", "Stop", etc. The two datasets provide a unique setup where digits 0-9 occur in different datasets but belong to the same semantic classes.

**EEG Signals** For our seizure analyses tasks, we use high-dimensional EEG data from different institutions and age groups: (a) Stanford Healthcare (SHC) with patients in the age group ~20-60 years, (b) Lucile Packard Children's Hospital (LPCH) with patients under 18 years and (c) publicly

available Temple University Hospital seizure corpus (TUH) (Shah et al., 2018; Obeid & Picone, 2016). 12 second or 60 second EEG clips with 19 channels, each sampled at 200Hz, are used for seizure detection.

**Vision** We use CIFAR-10 (InD), CIFAR-100 (Krizhevsky & Hinton, 2009) and SVHN (Netzer et al., 2011) datasets along with MNIST (Lecun et al., 1998) (InD), FashionMNIST (Xiao et al., 2017), eMNIST (Cohen et al., 2017) and kMNIST (Tarin et al., 2018) (collectively referred to as $x$-MNIST) datasets for image classification as is typically reported in literature (Hendrycks & Gimpel, 2017; Lee et al., 2018; Ren et al., 2019).

## B.2 BASELINES

We use the following baselines to compare against our framework. We don't compare with methods that require exposure to outliers to match our framework settings.

**MSP**: Hendrycks & Gimpel (2017) use the maximum softmax probability (MSP) $p(\hat{y}|x) = \max_k p(y = k|x)$ as the uncertainty score. OOD inputs tend to have lower MSP than InD data.

**Predictive Entropy**: Ren et al. (2019) show that using high entropy of the predicted class distribution $-\sum_k p(y = k|x) \log p(y = k|x)$ as an indicator for OOD inputs is a strong baseline.

**KL-divergence with Uniform distribution**: Hendrycks et al. (2019a) use the KL divergence of the softmax predictions to the uniform distribution $U$, $\text{KL}(U||p(y|x))$ to identify OOD inputs.

**ODIN**: Liang et al. (2018) use temperature-scaling and input perturbations to increase the gap between OOD and InD data using MSP. We fix $T = 1000, \epsilon = 1.4\text{e}^{-3}$, following their most common setting instead of tuning the parameters with outlier exposure.

**Vanilla Mahalanobis**: Lee et al. (2018) use the Mahalanobis distance from the nearest class-conditional Gaussian with shared covariance as the uncertainty score. This approach directly fits in with our spatial scoring method (Section 3.2.1) though we use class-wise covariance matrices.

**Test-time Dropout**: Gal & Ghahramani (2016) show that Monte Carlo (test-time) dropout to estimate the prediction distribution give good uncertainty estimates. Note that it is computationally intensive requiring $k$ forward passes of the classifier, with $k$-fold increase in runtime. We use $k = 10$.

## B.3 MODEL TRAINING & HYPERPARAMETERS

**Image Encoders & Pre-processing** We carry out our vision experiments using CNN-based classifiers, i.e. the LeNet architecture (Lecun et al., 1998) when working with $x$-MNIST data and the ResNet18 architecture (He et al., 2016) when working with CIFAR10/100/SVHN data. With the $x$-MNIST data, we normalize the inputs using the mean and standard deviation calculated from the training set for both training and inference, as is standard. With the CIFAR10/100/SVHN data, we apply the normalization transform along with augmentation strategies (random crop and horizontal flips) during training and only normalization during inference.

**Audio Encoders & Pre-processing** For our audio experiments, we use the M5 classifier (Dai et al., 2016) as our encoder. We also train a simple two-layer network – a static, normalized, log-scattering layer that extracts scattering coefficients from audio signals (Andreux et al., 2018) followed by a log-softmax layer that generates output probabilities to compare encoder capabilities (Section 5.4). Both models are trained only on classes 0-9 from the GSC dataset with the rest of the classes as OOD inputs. In all cases, we learn embeddings directly from raw, one second audio clips (resampled to 8kHz) without making use of any spectrogram features like MFCC (Xu et al., 2005).

**EEG Encoders & Pre-processing** For our seizure detection task, we use the Dense-Inception based models (Saab et al., 2020) as encoders. We use the same data pre-processing strategy as given by Saab et al. (2020). The models are trained on 19 channel, 12 second or 60 second raw EEG clips from InD datasets, resampled to 200Hz.

**Additional Details** We train all encoders on InD training data using standard weight initializations, SGD/Adam optimizer and ReLU non-linearities. We train and tune hyperparameters for our encoders using only InD data samples. We don't expose any outlier data to our encoders. We extract activations from a fully connected (FC) hidden layer before the logits layer to form latent space representations.

We normalize our distance scores and similarity scores before forming the spatial scores in cases where one of them is tightly bound while the other is not. For our temporal uncertainty scoring, we set window sizes $w_A$, $w_B = 25$ for vision tasks and $w_A$, $w_B = 50$ for audio and EEG tasks based on hyper-parameter tuning. We choose the reference window by sampling randomly from the training set (or coreset) and using its spatial scores in any order. We randomly shuffle the evaluation set and send in data points sequentially. We first obtain the spatial uncertainty scores for the evaluation set forming a 1D evaluation sequence, on which we form the sliding window sequentially. We choose the non-parametric, two-sided Mann-Whitney test (Mann & Whitney, 1947) at a significance level of $0.05$ as our measure $\mathcal{F}$. We perform two-centered kMeans clustering (MacQueen, 1967) on the temporal score sequence to identify uncertain model predictions and reject corresponding test samples as OOD.

## C  SEQUENTIAL EVALUATION: VISUALIZATIONS

Some examples of data streams generated from sampling EEG, audio and vision datasets are shown below (Figs. 2, 3, 4, 5, 6, 7, 8 and 9). The The X axis indicates the time sequence. The Y axis gives the uncertainty scores (Mann Whitney Scores are the same as TIME-LAPSE temporal scores). Using a two cluster KMeans algorithm, change points are detected and individual samples are accepted or rejected as InD.

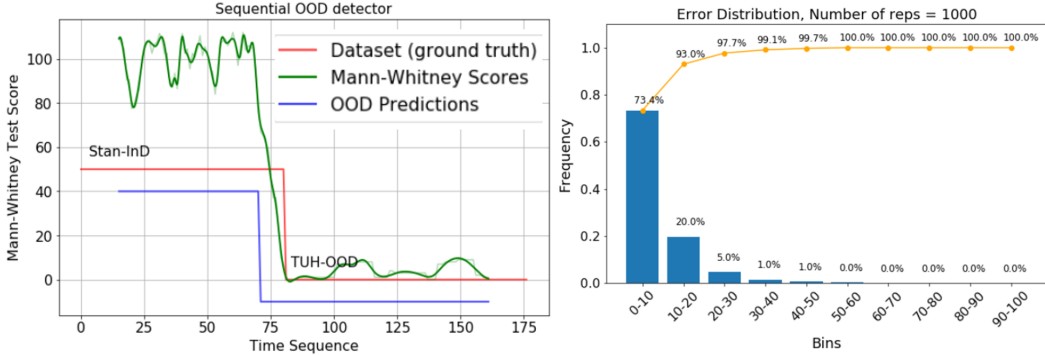

Figure 2: (left) Snapshot of the TIME-LAPSE temporal scores. Within a window of 15 samples, it is able to detect the change point for the seizure detection task. (right) % Error distribution over 1000 trials.

## D  ABLATIONS: IMPORTANCE OF INDIVIDUAL SCORES IN TIME-LAPSE

We study the effect of individual scores (distance spatial score vs similarity spatial score, spatial scores vs temporal scores, just using temporal scores) in this section. We also compare the effect of having shared covariance assumption in the Mahalanobis distance score (baseline Vanilla Mahalanobis) as opposed to using the separate covariance assumption that we use in our Mahalanobis distance score. From Table 8, we see that:

- Shared vs Separate covariances: We see that using the shared covariance assumption is more restrictive and does not fit various types of data. From Table 8, we see that it performs worse across all tasks and domains in comparison to the more relaxed separate covariance assumption we make for our Mahanalobis distance formulation (Section 3.2.1).

- Only distance scores: It is easy to see how the distance only scores perform in comparison to TIME-LAPSE. They follow the same trend as TIME-LAPSE regarding semantic overlap, i.e., they are more sensitive to semantic overlap than dataset statistics. But their performance is not high enough to be competitive.

- Only similarity scores: We see that similarity scores are more tuned to direct similarity, whether they are dataset-based or semantic-based. TIME-LAPSE allows us to get the benefits of both our distance scores and similarity scores with a competitive boost in performance across all semantic tasks.

- Spatial vs Spatio-temporal: We provide a qualitative ablation of using just spatial scores versus using the added advantage provided by temporal scoring through Figs. 3, 4, 5. We see from the plots that the margin between uncertainty scores of InD and OOD data is low for any of the baselines and the purely spatial TIME-LAPSE scores. Addition of the temporal scores greatly increases the separability between them leading to an increase in performance not only as measured by net AUROC but also increased interpretability.

- Only temporal (directly on embeddings): Conceptually, it is possible to use our temporal scoring approach directly on the (non-scalar) latent-space embeddings, skipping the spatial scoring step entirely. However, we have evidence from literature (Ramdas et al., 2015) to show that that hypothesis based tests such as our temporal approach do not scale well to the multivariate case. Hence, we do not consider it a viable option.

Table 8: Ablations showing the importance of individual scores on TIME-LAPSE. Semantic, Dataset, All refer to modes of evaluation. Semantic (audio) involves considering spoken digits from different datasets to be InD. Dataset (audio) involves considering spoken digitis from different datasets to be OOD. Semantic (MNIST) involves removal of e-MNIST classes that share ambiguous semantic overlap {o,l,i,z,y,s,q} with digits 0-9. All (MNIST) involves considering them as OOD along with other classes.

|  | Audio | | CIFAR | | MNIST | |
| --- | --- | --- | --- | --- | --- | --- |
|  | Semantic | Dataset | SVHN | CIFAR100 | Semantic | All |
| Only distance (shared covariance) | 0.658 | 0.480 | 0.962 | 0.694 | 0.951 | 0.919 |
| Only distance (ours) (separate covariance) | 0.728 | 0.502 | 0.980 | 0.832 | 0.962 | 0.948 |
| Only similarity (ours) | 0.658 | 0.817 | 0.984 | 0.880 | 0.989 | 0.970 |
| TIME-LAPSE (ours) | 0.739 | 0.506 | 0.989 | 0.875 | 0.990 | 0.974 |

## E  ABLATIONS: CORESET SIZE

TIME-LAPSE's spatial score uses the cosine similarity metric to compare the similarity of a test sample's latent-space embeddings to those of the training data pairwise. Potentially, this could be a computationally intensive process, especially for large dataset sizes. We reduce the computation costs, memory overhead and latency by using a subset of training data (coreset) instead of the full training set. The coreset is extracted by randomly sampling a fraction from every class in the training data. We show that in case of a large dataset, a small coreset will be sufficient to achieve a similar performance for downstream tasks. We can see from Table 9 that just 2% to 10% of the training dataset is sufficient to achieve a similar performance as using the full training dataset. In fact, for our clinical EEG dataset, we use just 2% of the training samples for all spatial score calculations and reach the state-of-the-art. This encouraging result also paves the way for choosing different sampling strategies that can ensure more "representational" samples (based on domain knowledge) from the training set are included in the coreset as part of future work. We note that similar trends with coresets are observed by Tack et al. (2020) though they use a different strategy to select their coreset.

Table 9: Coreset Ablations. AUROC for OOD detection for various coreset sizes (% of training samples). Even 2% and 10% of training data give good results.

| Coreset (%) | Audio | CIFAR | MNIST | EEG |
| --- | --- | --- | --- | --- |
| 1% | 0.7360 | 0.8543 | 0.9690 | - |
| 2% | 0.7385 | 0.8581 | 0.9702 | 0.771 |
| 10% | 0.7390 | 0.8620 | 0.9710 | - |
| 100% | 0.7399 | 0.8663 | 0.9721 | - |

## F  ABLATIONS: EFFECT OF ENCODER CAPACITY (SUPPLEMENTARY FIGURES)

We visualize the distributions of TIME-LAPSE spatial scores generated by two encoder architectures (Kymatio and M5) with varying modelling capacity. We see from Fig. 10 that more modelling capacity an encoder has, the better it is able to separate InD and OOD samples in its latent space. Note that in Fig. 10, the left plot shows the distribution of scores for InD samples from different datasets in blue and orange and that of all the OOD samples together in green. In the right plot,

however, blue indicate the merged InD samples from both datasets while orange represents class-wise distributions across OOD samples.

In Fig. 11, we keep the architecture of the encoder fixed, i.e., Kymatio, and vary how well it is trained. We use two encoders of the same architecture, train one on GSC 0-9 classes and another on FSDD 0-9 classes. Both the encoders are thus trained on the same InD classes but different datasets. The model trained on FSDD is able to reach high accuracy on the FSDD test sets whereas the one trained on GSC cannot. From the plots, we see that the model trained on FSDD actually overfits to the data since it is able to distinguish between InD classes from different datasets (Fig. 11, right).

## G   QUALITATIVE ANALYSES

We plot the UMAP (McInnes et al., 2018) embeddings of our raw InD data and their extracted latent space representations from the trained encoders along with those of OOD data. Fig. 12 shows the plots for our audio GSC dataset. We see that the raw data are too high-dimensional to be captured according to their class labels by UMAP. On the other hand, the extracted representations arrange themselves in well-separable clusters showing the representational capacity of the trained encoder. Most distance-based OOD detectors assume OOD representations will get mapped far away from the InD representations in the latent-space. This is a good assumption and holds in many cases. Indeed, the spatial scorer in TIME-LAPSE uses this hypothesis (along with others) to derive its predictive uncertainty score. However, OOD inputs can get mapped very close to InD inputs in the representational space under most distance metrics. Fig. 12 (right) shows the overlay of OOD inputs in the latent-space. Although it is just a 2D projection and the extracted representations have higher dimensions, this figure still shows that OOD inputs can get mapped near to InD inputs. In that case, purely distance-based spatial scores lose their power. TIME-LAPSE looks at multiple hypotheses to determine if samples are OOD. We find that the combination of our similarity score and distance score outperform each score individually. Moreover, since TIME-LAPSE has a temporal component to it, the temporal scorer acts as an additional constraint to ensure OOD inputs are separable. Thus, within a few samples, TIME-LAPSE can indicate when samples are going out-of-distribution with much better accuracy.

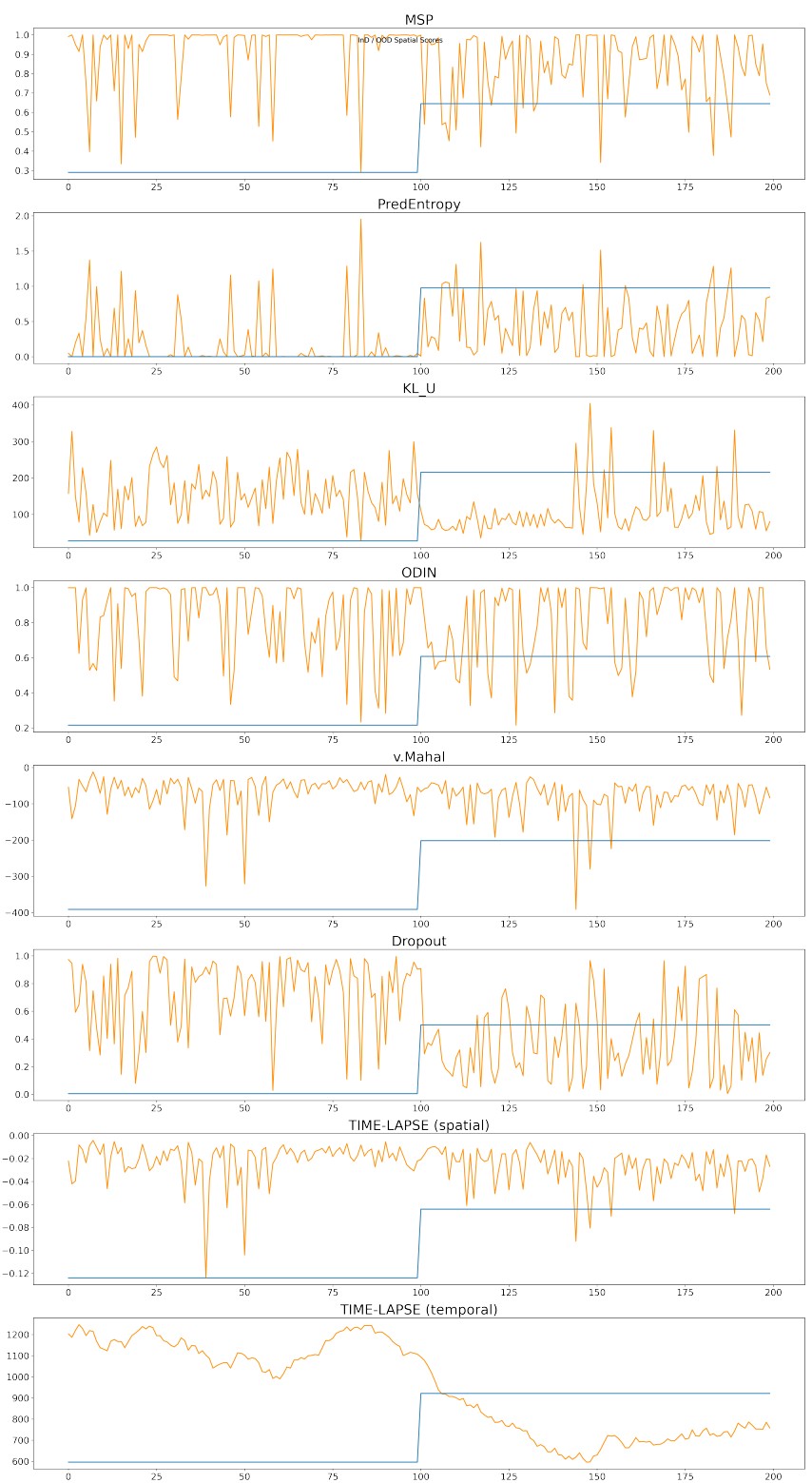

Figure 3: Uncertainty scores (audio). TIME-LAPSE (temporal) shows the most separability between InD (blue low segment) and OOD (blue high segment)

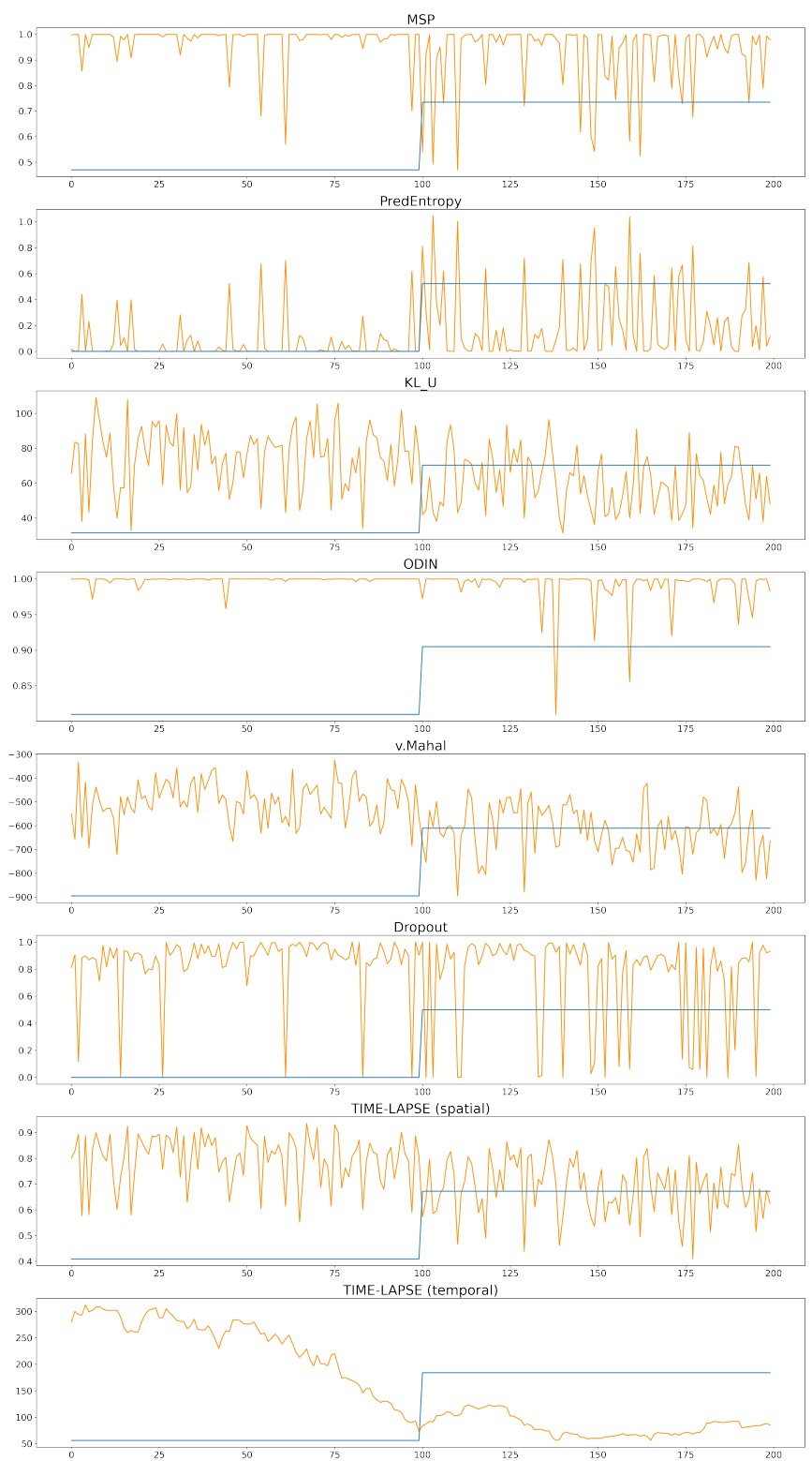

Figure 4: Uncertainty scores (CIFAR10). TIME-LAPSE (temporal) shows the most separability between InD (blue low segment) and OOD (blue high segment)

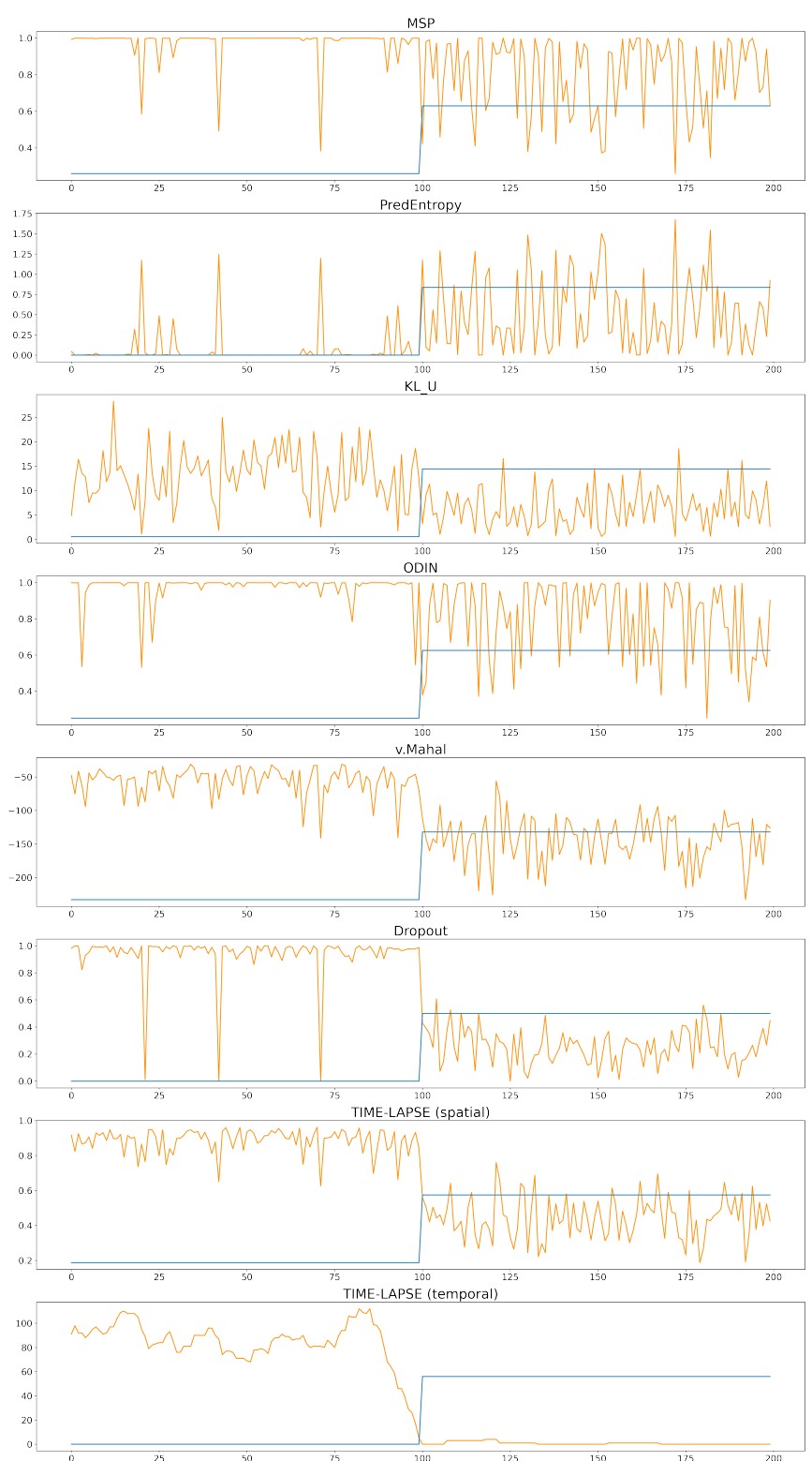

Figure 5: Uncertainty scores (MNIST). TIME-LAPSE (temporal) shows the most separability between InD (blue low segment) and OOD (blue high segment)

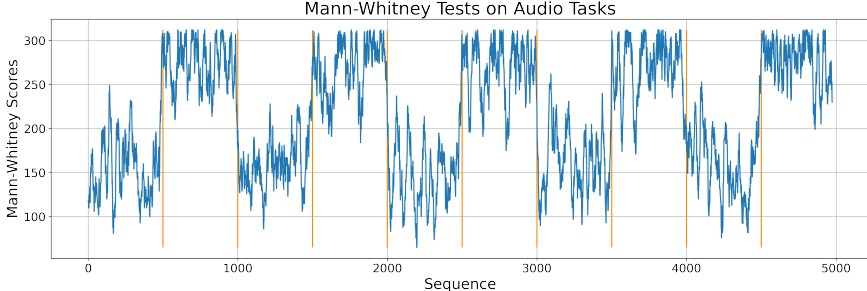

Figure 6: Sequential Evaluation: Data stream with distribution shifts occurring every 500 steps. The orange lines indicate true change points. Blue plot represents the Mann-Whitney scores generated from the spatial scores of audio clips.

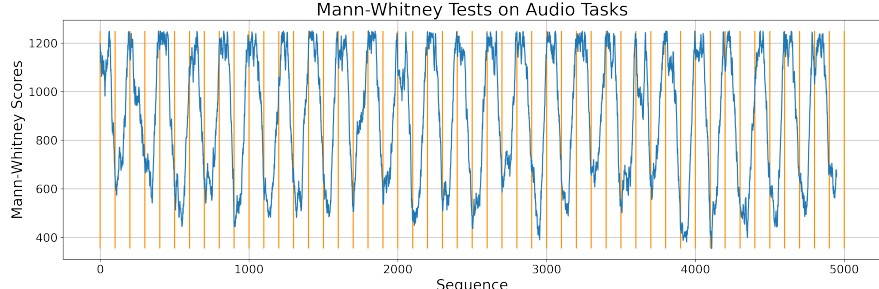

Figure 7: Sequential Evaluation: Data stream with distribution shifts occurring every 100 steps. The orange lines indicate true change points. Blue plot represents the Mann-Whitney scores generated from the spatial scores of audio clips.

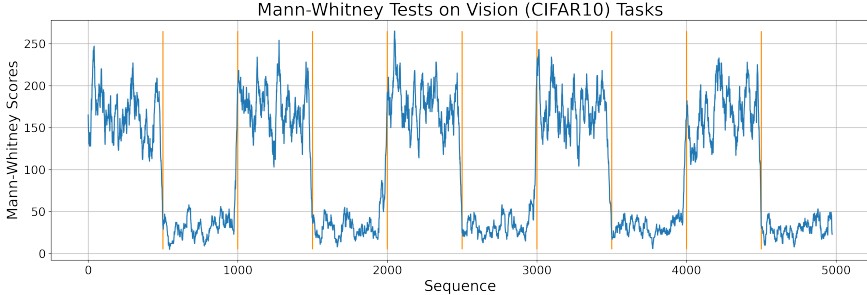

Figure 8: Sequential Evaluation: Data stream with distribution shifts occurring every 500 steps. The orange lines indicate true change points. Blue plot represents the Mann-Whitney scores generated from the spatial scores of CIFAR + OOD images.

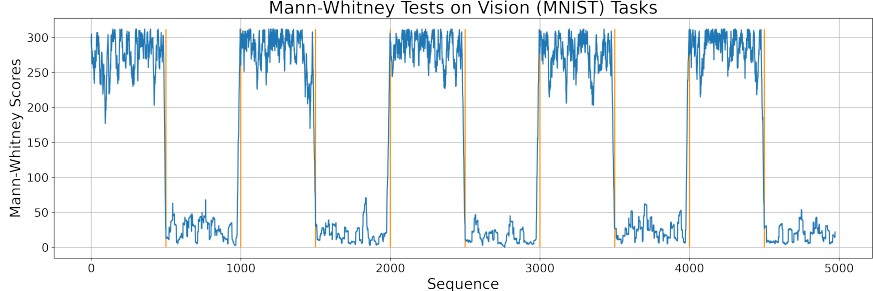

Figure 9: Sequential Evaluation: Data stream with distribution shifts occurring every 500 steps. The orange lines indicate true change points. Blue plot represents the Mann-Whitney scores generated from the spatial scores of MNIST + OOD images.

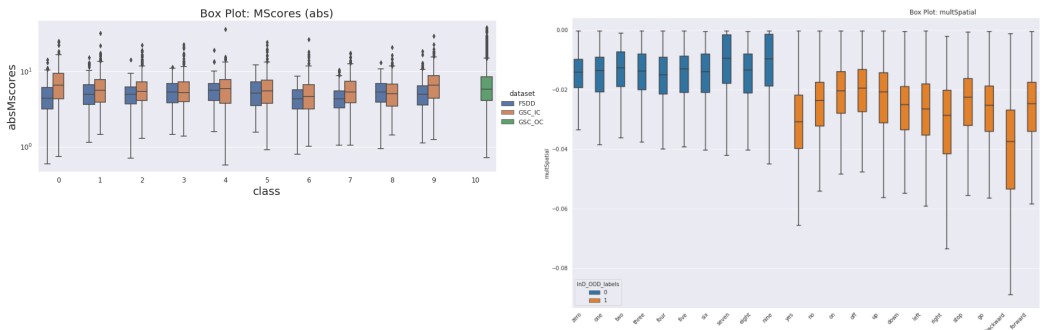

Figure 10: Distribution of TIME-LAPSE scores across semantic InD (GSC 0-9, FSDD 0-9) and OOD classes with (left) Kymatio encoder, (right) M5 encoder, class-wise OOD scores are shown. GSC 0-9 and FSDD 0-9 are merged here and only part of the full list of OOD classes shown. All GSC OOD classes are merged together as green in the left figure.

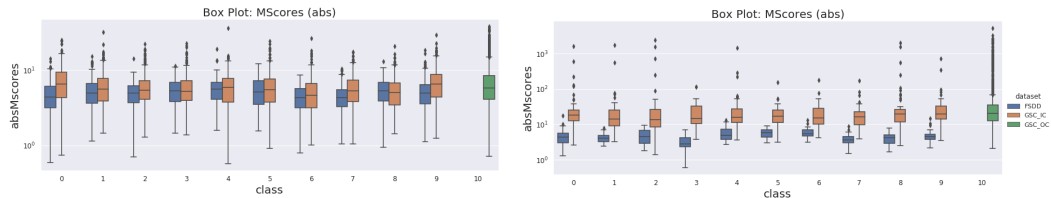

Figure 11: Distribution of TIME-LAPSE scores with trained Kymatio encoder. (left) Kymatio encoder trained with GSC 0-9 data, achieving 0.75 classification accuracy. The InD scores from the two datasets (blue and orange) are indistinguishable (right) Kymatio encoder trained with FSDD 0-9 data, achieving 0.97 classification accuracy. The InD scores between the two datasets are separable, showing that it has overfit to the dataset statistics.

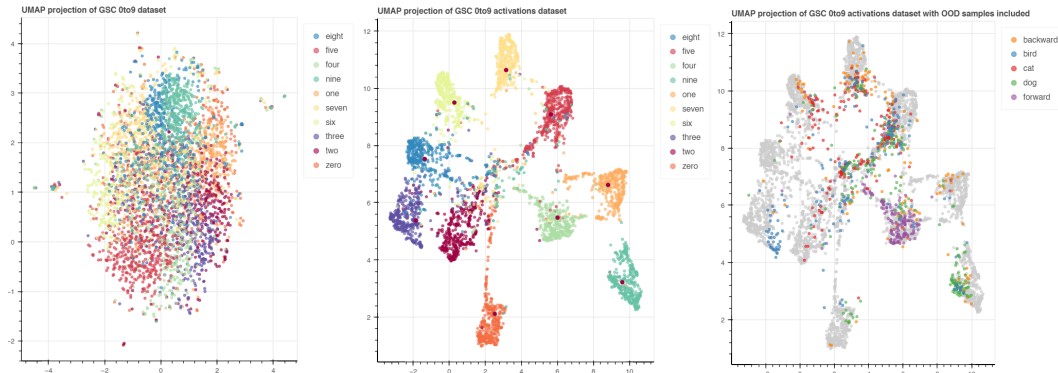

Figure 12: UMAP embeddings of GSC data (left) raw InD data (middle) representations from the trained encoder, i.e. a 2D projection of the latent space (c) OOD samples added to the mix (only few classes are shown for visualization purposes).

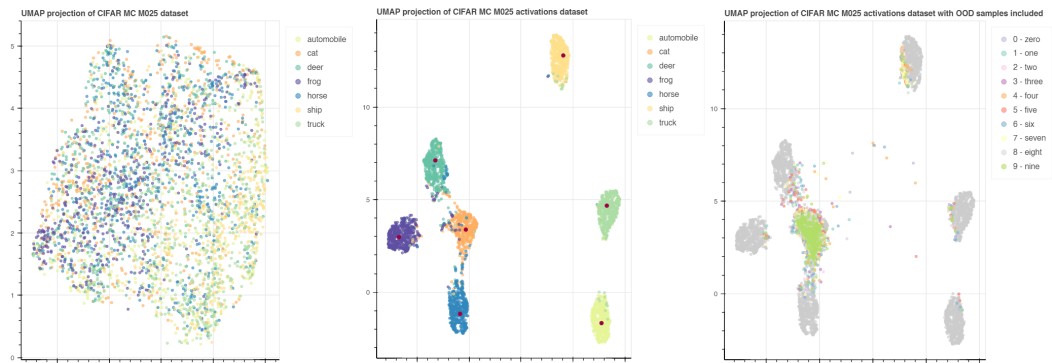

Figure 13: UMAP embeddings of CIFAR data (left) raw InD data (middle) representations from the trained encoder, i.e. a 2D projection of the latent space (c) OOD samples added to the mix (only few classes are shown for visualization purposes).

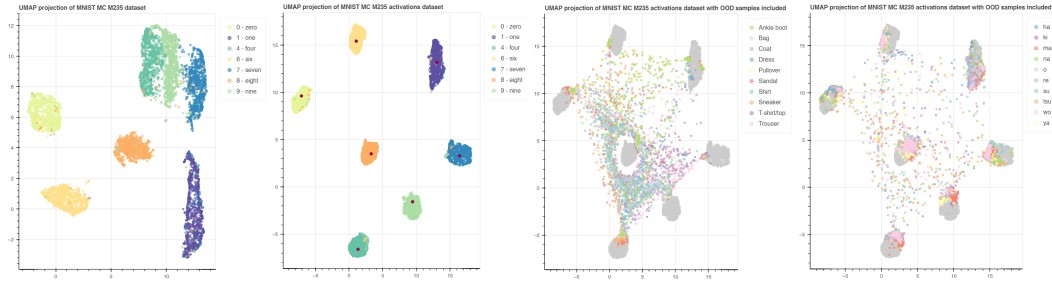

Figure 14: UMAP embeddings of MNIST data (left) raw InD data (middle) representations from the trained encoder, i.e. a 2D projection of the latent space (c) OOD samples added to the mix (only few classes are shown for visualization purposes).

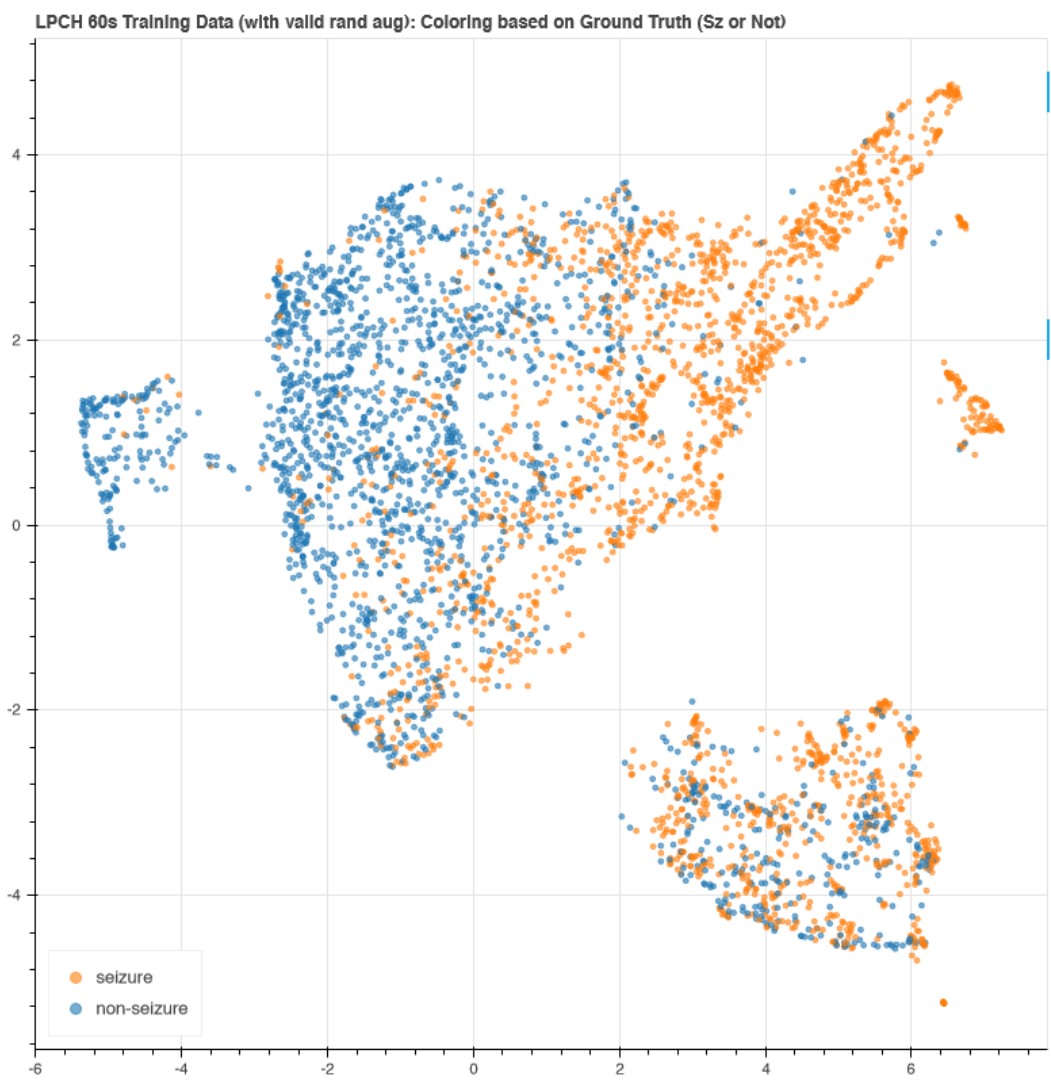

Figure 15: UMAP embeddings of EEG data (left) raw InD data (middle) representations from the trained encoder, i.e. a 2D projection of the latent space (c) OOD samples added to the mix (only few classes are shown for visualization purposes).

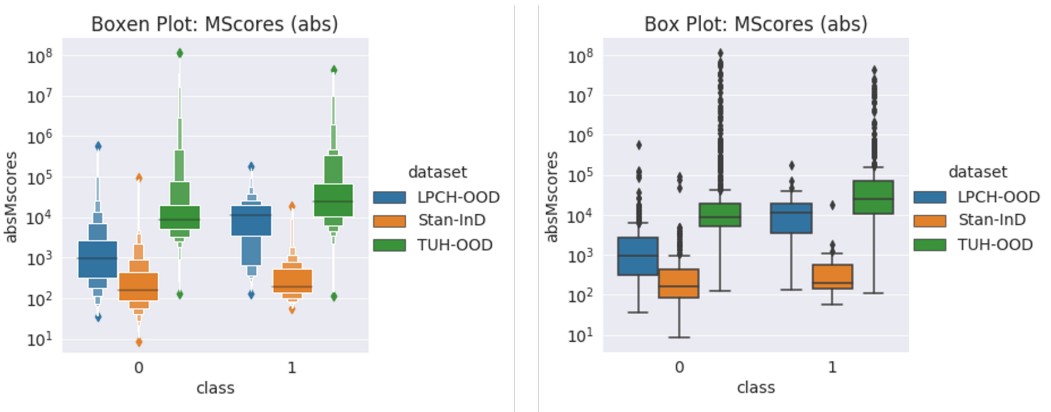

Figure 16: TIME-LAPSE score distribution on EEG data

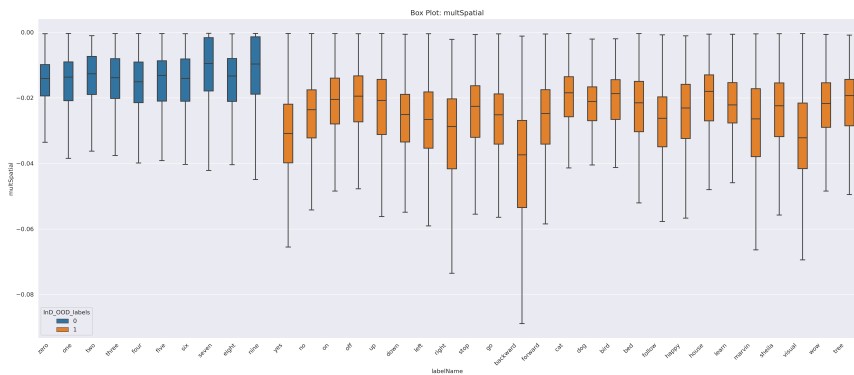

Figure 17: TIME-LAPSE score distribution on GSC

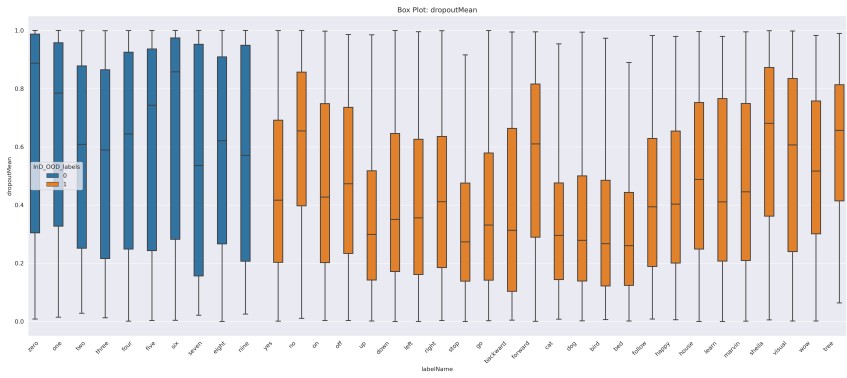

Figure 18: Test-time Dropout score distribution on GSC

