# OpenReview forum: "TIME-LAPSE: Learning to say “I don't know” through spatio-temporal uncertainty scoring"
_ICLR.cc/2022/Conference — ICLR 2022 Submitted_

### Official Review · Reviewer_qgxp · 2021-11-01

**Correctness:** 1
**Technical Novelty And Significance:** 2
**Empirical Novelty And Significance:** 2
**Recommendation:** 5
**Confidence:** 4

**Details Of Ethics Concerns:**

No comments

**Main Review:**

The strengths of this paper are:
+ It addresses a highly relevant and important problem, OOD.
+ It provides thorough experiments with baseline comparisons.
+ It is well organized and easy to read.

Unfortunately, the paper suffers from a number of major weaknesses:
- Firstly, the title and beginning of intro is misleading - the paper does not address uncertainty quantification or calibration, it just provides a method for OOD. Please narrow down the scope of the first intro paragraphs, and select a more focused title.
- Secondly, the terminology "spatial" and "temporal" is not correctly used. The spatial dimension would refer to dimensions within a data-point in time, such as the vertical and horizontal dimension in a video frame. You could instead talk about time-independent and time-dependent distance measure.
- Third, your temporal uncertainty bears large resemblance to filtering and temporal smoothness methods - Kalman and particle filters pre-deep learning, and deep learning methods such as LSTM and CNN over time. The similarity to these methods should be discussed. Moreover, you should refer to some work from the by now large literature on uncertainty estimation and calibration of classification models.
- Last, although rigorous, the experimental results lack standard deviations. Without this information, it is not possible the significance of the results presented in tables 1-3.

Given this, I recommend to not accept this paper, as it is not yet in a state to be published in ICLR. However, given the encouraging results and the importance of the problem, I encourage the authors to address the weaknesses above, revise the paper, and submit it to a later venue.

EDIT AFTER REBUTTAL: The authors have addressed my comments to a large extent, and the paper now looks much nicer - although, I am still not convinced about the technical novelty compared to the vast literature on filtering, outlier detection, uncertainty estimation and calibration. Hence, I raise my grade to 5.

**Summary Of The Paper:**

This paper presents a method for out-of-distribution (OOD) detection taking temporal consistency into account - both the position of a new data-point in the latent space in relation to the other observed data, as well as the relation to temporally close (recently observed) data are taken into account. Experiments are performed on both audio, video and EEG data, and comparisons are made against six baselines.

**Summary Of The Review:**

The strengths of this paper are that is addresses an important problem, it provides thorough experiments and it is well organized and easy to read.

Unfortunately, the paper suffers from a number of major weaknesses, in that the title and intro paragraphs are misleading, there are issues with terminology and there are missing references to the literature on filtering and uncertainty estimation. Moreover, the results do not show if the differences to baselines are significant.

Given this, I recommend to not accept this paper, but I encourage the authors to address the weaknesses and resubmit to a later venue.

EDIT AFTER REBUTTAL: The authors have addressed my comments to a large extent, and the paper now looks much nicer - although, I am still not convinced about the technical novelty compared to the vast literature on filtering, outlier detection, uncertainty estimation and calibration. Hence, I raise my grade to 5.

---

> ### Author Response · Authors · 2021-11-18
> **Standard deviations added, weaknesses addressed (manuscript: title, intro, related works updated)**
>
> We thank the reviewer for their valuable feedback & encouraging comments and for highlighting the strengths in our paper (relevance, thorough experimentation, well-organized & written). We have updated our manuscript to address your concerns and incorporated your suggestions. We give our detailed response below
>
> **_1. Although rigorous, the experimental results lack standard deviations. Without this information, it is not possible the significance of the results presented in tables 1-3._**
>
> Thank you for pointing this out. We have updated the manuscript to report means and standard deviations across 5 random runs to all our experiments and have verified that our results and conclusions remain the same. We report only means in the main manuscript due to our space constraint. But we report the expanded table with means and standard deviations in our Appendix (Tables 4,5,6 page numbers: 14,15). We hope that this will alleviate any major concerns about the significance of our results (that show substantial gains over baselines) and hope that the reviewer would be able to update their scores.
>
> **_2. The title and beginning of intro is misleading - the paper does not address uncertainty quantification or calibration, it just provides a method for OOD. Please narrow down the scope of the first intro paragraphs, and select a more focused title._**
>
> Thank you for the feedback here. We wish to clarify that the main aim of the paper is uncertainty estimation and we use the tasks of OOD detection and dataset shift detection as our means to validate our uncertainty estimates. To make this clear, we have chosen a more informative title “TIME-LAPSE: Uncertainty scoring via Latent-Space Embeddings over Time”. We have also revised our introduction to reflect that our goal is uncertainty estimation while OOD & drift detection are the concrete tasks to help us justify the quality of our uncertainty  estimates. Please let us know if you have any remaining concerns.
>
> **_3. The terminology "spatial" and "temporal" is not correctly used. The spatial dimension would refer to dimensions within a data-point in time, such as the vertical and horizontal dimension in a video frame. You could instead talk about time-independent and time-dependent distance measure._**
>
> Thank you for bringing this up and for the suggestion. We agree that this could be potentially confusing terminology. We would like to clarify that we are using the terms “spatial” and “temporal” in a different context here. Spatial in TIME-LAPSE refers to the **latent space** and not the space of the datapoint itself (as in the case of a video where space might refer to the dimensions within a video frame). Similarly, the temporal dimension in TIME-LAPSE is not time within the data point (as in a video or a time-series clip) but across **different data points**.
> That being said, we agree with you that the term “spatial-temporal” is largely overloaded with the video domain semantics. We have thus removed any explicit mention of the term “spatio-temporal” throughout the manuscript and instead use “latent-space embeddings over time” to remove any confusion. We still retain “spatial” scoring and “temporal” scoring since we believe they will not be taken out of context with our updated writing. Please let us know if you have further feedback.
>
> **_4. Your temporal uncertainty bears large resemblance to filtering and temporal smoothness methods - Kalman and particle filters pre-deep learning...  you should refer to some work from the … literature on uncertainty estimation and calibration of classification models._**
>
> Thanks for bringing this to our attention. We have updated our related works section (Section 2, page 3,4) to include these methods. If there are any specific papers that you believe will add value to our paper but we have missed, do let us know.
>
> We hope that we have resolved the reviewer’s concerns and hope that they can update their scores after going through our revisions. If there is any other feedback, we’d be glad to hear it.

---

> > ### Comment · Reviewer_qgxp · 2021-11-25
> > **Reply to authors**
> >
> > Thank you for your thorough rebuttal and revision! You have addressed my comments to a large extent, and the paper now looks much nicer - although, I am still not convinced about the technical novelty compared to the vast literature on filtering, outlier detection, uncertainty estimation and calibration. Hence, I raise my grade to 5.

---

> > > ### Author Response · Authors · 2021-11-26
> > > **Addressing novelty concerns (reply to reviewer)**
> > >
> > > We thank the reviewer for the positive comments and for updating their score.
> > >
> > > We’d like to address their concern about our work’s technical novelty below. Though the literature on outlier detection, uncertainty estimation and calibration is vast (we have summarized them in the paper, Section 2, page 3)  and this subject has been studied from different lenses over decades, there are clear novelties in our framework compared to previous and contemporaneous work:
> > >
> > > 1. The fundamental concept of TIME-LAPSE is completely new and unique. No other works have used **la**tent **sp**ace **e**mbeddings over **time** for uncertainty quantification. There have been approaches that are “spatio-temporal” in the video domain (and some time-series) but they do not share any technical details with our work. Apart from our initial terminology that used spatio-temporal (which we have removed to avoid confusion), there is no commonality with those types of approaches. We’d like to emphasize that our framework does not require the data to have any time dependency unlike methods that are called spatio-temporal.
> > >
> > > 2. In our related works section (Section 2, page 3), we go over the different types of uncertainty estimation methods (bayesian, MCMC dropout, likelihood-based, energy-based models), calibration (frequentist notion of uncertainty using proper scoring techniques), outlier detection and OOD detection methods (either classic 1D, time-series based, reconstruction based,  likelihood-based, energy-based, etc), all of which share no technical content with TIME-LAPSE. We use the Mahalanobis distance as one part of our framework (spatial distance score) as do Lee, et al, 2018  [1]. However, in [1], their entire method just uses the Mahalanobis distance (our baseline Vanilla Mahalanobis in our paper) whereas in TIME-LAPSE, we have 3 different components: the spatial distance score, the spatial similarity score and the temporal score that contribute to our performance. As we state in Section 3.2.1 (page 5), TIME-LAPSE will work with any appropriate choice of distance metric, and similarity metric.
> > >
> > > 3. Filtering and hypothesis testing based methods (as the reviewer pointed out) have been used for outlier detection in the past, but have solely been limited to 1D time-series data (speech, sensor, etc) [2, 3, 4, 5]. They have been shown to not scale well to the multi-dimensional case [6]. On the other hand, in our approach, we present an elegant way to combine the power of such methods without losing performance when using high-dimensional data.
> > >
> > > 4. We are the first to evaluate and benchmark six strong baselines apart from TIME-LAPSE on (i) OOD detection and (ii) drift detection, both of which are crucial for deployment and monitoring of ML models.
> > >
> > > 5. We are also the first to benchmark these six baselines and TIME-LAPSE on multiple, diverse domains (vision, audio and clinical EEGs) and measure their affinity for semantic content preservation. We are also the first to show uncertainty quantification for EEG-based seizure analyses tasks.
> > >
> > > 6. We show strong empirical novelty as TIME-LAPSE reaches the state-of-the-art (SOTA) in all the domains & over all tasks and shows tremendous affinity for semantic content compared to any of the other baselines.
> > >
> > > We hope that we have highlighted TIME-LAPSE’s unique strengths and technical novelties compared to prior work that sets it apart from other methods. If there are any other concerns from the reviewer, please do let us know. We’d be happy to hear it and address them.
> > >
> > >
> > >  [1] Kimin Lee, Kibok Lee, Honglak Lee, and Jinwoo Shin. A simple unified framework for detecting
> > > out-of-distribution samples and adversarial attacks. Advances in Neural Information Processing
> > > Systems (NeurIPS), 2018.
> > > [2] Rudolph Emil Kalman. A new approach to linear filtering and prediction problems. Transactions of the ASME–Journal of Basic Engineering, 1960.
> > > [3] Rudolph van der Merwe, Arnaud Doucet, Nando de Freitas, and Eric Wan. The unscented particle filter. Advances in Neural Information Processing Systems (NeurIPS), 2001
> > > [4] Daniel Kifer, Shai Ben-David, and Johannes Gehrke. Detecting change in data streams. International Conference on Very Large Data Bases (VLDB), 2004.
> > > [5] Samaneh Aminikhanghahi and Diane J. Cook. A survey of methods for time series change point detection. Knowledge and information systems, 51(2):339–367, 2017.
> > > [6] Aaditya Ramdas, Sashank J. Reddi, Barnabás Póczos, Aarti Singh, and Larry Wasserman. On the decreasing power of kernel and distance based nonparametric hypothesis tests in high dimensions. Proceedings of the Twenty-Ninth AAAI Conference on Artificial Intelligence (AAAI), 2015.

---

### Official Review · Reviewer_CrvH · 2021-11-06

**Correctness:** 2
**Technical Novelty And Significance:** 2
**Empirical Novelty And Significance:** 2
**Recommendation:** 3
**Confidence:** 3

**Main Review:**

*Strengths*:
* The idea of using temporal structure in data and coresets for OOD detection are both interesting ideas.
* The paper conducted experiments across several domains, involving a number of different datasets and tasks.

*Weaknesses*:
* This work is not the first spatio-temporal model for OOD detection as claimed. See [1] for a review. See [2, 3, 4] for examples of how other works deal with problems similar as ‘an obstacle detector deployed in a self-driving car will see images correlated over time.’ (which is an example used in the introduction) See [5] for an example of how the spatio-temporal structure is used when applied on EEG datasets. In light of these works and lack of discussion and comparisons, the novelty and contributions of the work are unclear.

* The method makes sense overall but lacks justification of certain choices. For instance, the choice of the Mahalanobis metric and Gaussian assumptions in OOD detection is unclear. What does the combined spatial uncertainty score imply? How is the coreset extracted and how do the number of points in the coreset impact detection? Some analysis of these different parameters and aspects of the method should be explored further. Because there are no strong theoretical justifications, ablations of some of the different components of the proposed model are necessary.

* How is the model used for image datasets (CIFAR)? The beginning of the results discusses how OvR classification can be used for examining OOD but it is unclear how these datasets are used in this case where the temporal structure of inliers is a core assumption. If different classes are presented as certain temporal rates, this information should be discussed and analyzed. How does the temporal rate of different classes impact learning or detection?

* How is the reference window and the sliding window selected? Does setting this parameter involve an estimation of how temporally distributed OOD examples will be? Details on this and other choices / hyperparameters should be provided.

Minor comments:
- The authors claim that their proposed method is more robust to dataset statistics and focuses on the semantic meaning of the data. However, the same was not discussed for the EEG experiments, where the InD/OOD is decided based on either patient population or different patient demographics. It would be interesting to see this discussion.
- Fix the equation in 3.1: ‘th’ as threshold should be explained.

[1] Abdar, M., Pourpanah, F., Hussain, S., Rezazadegan, D., Liu, L., Ghavamzadeh, M., ... & Nahavandi, S. (2021). A review of uncertainty quantification in deep learning: Techniques, applications and challenges. Information Fusion.
[2] Zhao, Y., Deng, B., Shen, C., Liu, Y., Lu, H., & Hua, X. S. (2017, October). Spatio-temporal autoencoder for video anomaly detection. In Proceedings of the 25th ACM international conference on Multimedia (pp. 1933-1941).
[3] Nguyen, T. N., & Meunier, J. (2019). Anomaly detection in video sequence with appearance-motion correspondence. In Proceedings of the IEEE/CVF International Conference on Computer Vision (pp. 1273-1283).
[4] Zhou, J. T., Du, J., Zhu, H., Peng, X., Liu, Y., & Goh, R. S. M. (2019). Anomalynet: An anomaly detection network for video surveillance. IEEE Transactions on Information Forensics and Security, 14(10), 2537-2550.
[5] Fernando, T., Denman, S., Ahmedt-Aristizabal, D., Sridharan, S., Laurens, K. R., Johnston, P., & Fookes, C. (2020). Neural memory plasticity for medical anomaly detection. Neural Networks, 127, 67-81.
[6] Hendrycks, D., Mazeika, M., & Dietterich, T. (2018). Deep anomaly detection with outlier exposure. arXiv preprint arXiv:1812.04606.


**Summary Of The Paper:**

This paper proposes a spatio-temporal training method to quantify uncertainty and detect out-of-distribution samples. The authors apply their method to audio speech classification, seizure detection using EEGs, and image classification.

**Summary Of The Review:**

The idea of using temporal structure to perform out-of-distribution detection is an interesting problem with many important applications. While the authors provide some experimental results that suggest their approach outperforms other methods, there was little justification for the different choices made and no analysis of how these decisions impact their model's performance. In addition, the authors did not discuss or compare other approaches for estimating spatiotemporal uncertainty from sequential data and thus it is difficulty to assess the novelty and contributions of the work.

---

> ### Author Response · Authors · 2021-11-19
> **Terminology clarification, updated experiment details + design choices (1/2)**
>
>
> We thank the reviewer for their insightful comments & feedback and for highlighting our strengths (temporality, diversity of datasets, tasks) and finding our idea & methodology interesting. We give the responses to specific comments below
>
> **_1. This work is not the first spatio-temporal model for OOD detection as claimed. See [1] for a review. See [2, 3, 4] for examples of how other works deal with problems similar as ‘an obstacle detector deployed in a self-driving car will see images correlated over time.’ (which is an example used in the introduction) See [5] for an example of how the spatio-temporal structure is used when applied on EEG datasets. In light of these works and lack of discussion and comparisons, the novelty and contributions of the work are unclear._**
>
> Thank you for the feedback!. However, we believe there are 2 misunderstandings about our work compared to the references cited here.
>
> a. We’d like to clarify that we’re using the terms “spatial” and “temporal” in a different context compared to the video community. Spatial in TIME-LAPSE refers to the **latent space** and not the space of the datapoint itself (as in a video where space refers to the dimensions within a video frame). Similarly, temporal in TIME-LAPSE is not time within the data point (as in a video or a time-series clip) but across **different data points**. We believe the term “spatial-temporal” could be the origin for this misunderstanding and apologise for using overloaded terminology. We have updated our paper to remove that term to avoid confusion with the video community. Please let us know if there are any lingering questions / concerns regarding this point after reading our updated intro.
>
> b. The goal of TIME-LAPSE is uncertainty quantification associated with model predictions, which we evaluate using distributionally shifted samples under OOD detection and drift detection tasks. However, the references provided [1,2,3,4] are doing *video anomaly detection* which deals with detection of anomalous or unexpected **events**  within a video. Similarly, ref [5] which does *EEG anomaly detection* again deals with detection of anomalous **events** such as seizures from normal EEGs, i.e., it is trained on normal EEG data and is trained to identify anomalies from normal EEGs. To use the same example, in contrast, our in-distribution EEG data has both normal EEGs and seizures. Our network is doing a binary classification for seizure detection and TIME-LAPSE scores indicate when the model predictions should be uncertain (when it encounters a new seizure type, significant changes in distribution of the signals, neonates, etc). In light of this, we do believe we are the first to show deep learning based uncertainty quantification for EEG seizure detection tasks.
>
> We hope that this has clarified the differences between our work and the references provided and highlighted our work’s novelty and contributions.
>
> **_2. The choice of the Mahalanobis metric and Gaussian assumptions in OOD detection is unclear._**
>
> Thank you for the comment. Mahalanobis distance and the multivariate class-conditional Gaussian assumption (Gaussian Discriminant analysis) have been well-studied and shown to outperform Euclidean distance (Lee, et al. A Simple Unified Framework for Detecting OOD Samples and Adversarial Attacks, NeurIPS 2018). This held true in preliminary experiments with our data
>
> **_3. What does the combined spatial uncertainty score imply?_**
>
> Thank you for the question. The distance score (Mahalanobis) estimates how “far” a sample is from other known InD samples in the latent space while the sim score (cosine-similarity) estimates how similar it is to other known InD samples. We found empirically that both scores perform worse individually than the combined spatial score
>
> **_4. How is the coreset extracted and how do the number of points in the coreset impact detection? Some analysis of these different parameters and aspects of the method should be explored further._**
>
> Thanks for the question. The coreset is extracted by randomly sampling a fraction from every class in the training data. For the large EEG dataset, we use only 2% of the training data for our results. We also plan to add a section to the Appendix with an ablation on coreset sizes for other datasets as well, but we hope that our success with just 2% of the large EEG training set will alleviate the reviewer’s concern.
>
> _(**Updates:** We have carried out the experiments and have added our results and description to Appendix Section E (page 18). We are happy to report that the performance is similarly high even when using only 2% or 10% of the training data and believe that it’s another strength of our method that can now be highlighted.)_

---

> > ### Author Response · Authors · 2021-11-19
> > **Terminology clarification, updated experiment details + design choices (2/2)**
> >
> > **_5. How is the model used for image datasets (CIFAR)? … If different classes are presented as certain temporal rates, this information should be discussed and analyzed. How does the temporal rate of different classes impact learning or detection?_**
> > **_6. How is the reference window and the sliding window selected? Does setting this parameter involve an estimation of how temporally distributed OOD examples will be? Details on this and other choices / hyperparameters should be provided._**
> >
> > Thank you for the question. For all OOD tasks (not just images), we choose the reference window by sampling randomly from the training set and using its spatial scores in any order. We randomly shuffle the evaluation set and send in data points sequentially. We first obtain the spatial uncertainty scores for the evaluation set forming a 1D evaluation sequence, on which we form the sliding window sequentially to run our temporal method to get final scores. We set window sizes based on hyper-parameter tuning. Updated Appendix B.3 with these details. Newly added figs. 3, 4, 5 may be useful for understanding this better. In our drift detection task (Sec 5.3), we generate data streams of length 10,000 samples from eval sets, changing the distribution every k samples in each trial (1000 trials, k = 50, 100, 200, 500, 1000, 5000)
> >
> > **_7. semantics of EEG_**
> >
> > That’s a great point. We believe adding this discussion may be out of scope for this paper (we are working on a follow-up study) but we observe interesting semantic effects in the EEG case as well. Example: seizure types unseen by the network will generate higher uncertainty scores. We added this line to the paper as well.
> >
> > **_8. equation in 3.1: ‘th’ as threshold should be explained_**
> >
> > Thank you for pointing it out! We have updated the equation.
> >
> > We hope that we have resolved the reviewer’s concerns and hope that they can update their scores after going through our revisions. If there is any other feedback or more questions, we’d be glad to hear it.

---

### Official Review · Reviewer_o7sa · 2021-11-07

**Correctness:** 3
**Technical Novelty And Significance:** 2
**Empirical Novelty And Significance:** 3
**Recommendation:** 5
**Confidence:** 4

**Main Review:**

The paper achieves state-of-the-art results on diverse established datasets and performance metrics. More importantly, the suggested framework appears to be more semantics-preserving than existing scores that rely on data-statistics.
However, I feel that some hypothesis are not very well corroborated (experimentally and/or theoretically) while some details are missing.
In particular,

(1) could the authors please provide more details on the embeddings used? Are they deterministic or stochastic? If they are deterministic, how are they formed exactly ( it is the concatenation of the output of the hidden layers of the networks?)

(2) the hypothesis stated in 3.2.1 (that in the latent space ood samples will lie in distance from  iid samples), especially for high-dimensional inputs. may not always hold. A visualization on cifar10 data will better support this claim. Do the authors think that modifying the objective function to explicitly support this claim could further improve performance? In particular, I am pointing the authors to the following work (for stochastic features), that may want to consider for further improving the performance:

[1] Sinha S, Dieng AB. Consistency Regularization for Variational Auto-Encoders. arXiv preprint arXiv:2105.14859. 2021 May 31.

(3) Moreover, what if the embeddings of iid form clusters (also appart from each other)? How many training samples would be needed in this case to sufficiently populate the coreset??  The computation of the similarity score also seems prohibitive for large-scale datasets that may need large coreset. I think it would be good if the authors could report an ablation study on the size of the coreset.

(4) I also think the authors should extent the ablation such that they report results i) based on the spatial scores only (only distance, only similarity, both) and/or ii) the temporal score (computed directly on the embedding h) to better demonstrate the importance of each score used

(5) results on large-scale datasets (imagenet) could strengthen the impact of the paper.

**Summary Of The Paper:**

The paper proposes a spatio-temporal approach for unsupervised out-of-distribution detection. In particular, it suggests a 'hierarchical' evaluation method based on computation of existing distance/similarity scores on hidden embeddings of high-dimensional data succeeded by a temporal component, that is also based on existing work, that treats chunks of  scores, computed by the first step, of input data as a stream. The suggested pipeline achieves state-of-the-art results on datasets coming from different domains ( image, audio, clinical data). More importantly, it demonstrates the importance of semantic context when deciding whether a sample is ood exhibiting better performance than existing baselines when semantic overlap should be taken into consideration.

**Summary Of The Review:**

The paper shows competitive results compared to strong baselines and a promising direction for semantic preserving out-of-distribution detection. However, I think some suggestions on the ablation study and some comments on the scalability on large datasets should be addressed to strengthen the paper.

---

> ### Author Response · Authors · 2021-11-19
> **Complexity concerns addressed, ablations comments explained (1/2)**
>
> We thank the reviewer for their valuable feedback & comments and for finding our method & direction promising (semantics preserving and SOTA)! We provide responses to individual points below
>
> **_1. Could the authors please provide more details on the embeddings used? Are they deterministic or stochastic? If they are deterministic, how are they formed exactly ( it is the concatenation of the output of the hidden layers of the networks?)_**
>
> Thank you for the question. We provide this detail in Appendix B.3. “We extract activations from a fully connected (FC) hidden layer before the logits layer to form latent space representations”.
>
> **_2. The hypothesis stated in 3.2.1 (that in the latent space ood samples will lie in distance from iid samples), especially for high-dimensional inputs. may not always hold. A visualization on cifar10 data will better support this claim._**
>
> Thank you for bringing this up. Absolutely. We provide a 2D visualization of our datasets & their embeddings in Appendix E:  Qualitative Analyses Figs. 12-15. We use the reviewer’s same point in the section as a justification - it is true that the first hypothesis works well but may not always hold true. But since TIME-LAPSE looks at multiple hypotheses to determine if samples are OOD, our similarity hypothesis and temporal correlation hypothesis boost our performance compared to other methods.
>
> **_3.Do the authors think that modifying the objective function to explicitly support this claim could further improve performance? In particular, I am pointing the authors to the following work (for stochastic features), they may want to consider for further improving the performance: [1] Sinha S, Dieng AB. Consistency Regularization for Variational Auto-Encoders. arXiv preprint arXiv:2105.14859. 2021 May 31._**
>
> We thank the reviewer for the suggestion. Modifying the objective to support this is definitely an interesting idea and worth exploring in a separate study as part of our future work. While TIME-LAPSE can easily accommodate and benefit from models trained with this objective, since it requires retraining the network it is not aligned with the scope of our paper. This paper’s objective is to be able to quantify predictive uncertainties for model predictions during inference, irrespective of the way it has been trained.
>
> **_4. What if the embeddings of iid form clusters (also apart from each other)? How many training samples would be needed in this case to sufficiently populate the coreset?? The computation of the similarity score also seems prohibitive for large-scale datasets that may need a large coreset. I think it would be good if the authors could report an ablation study on the size of the coreset._**
>
> The reviewer raises good points. Firstly, we expect InD embeddings to form separable clusters based on their classes and this is true for our datasets (Figs. 12-15). This will, in fact, be beneficial for TIME-LAPSE’s performance, since both our distance score and similarity score work based on comparisons with each class cluster.
>
> Secondly, the above point does not determine how many training samples are needed to populate the coreset. The coreset is extracted by randomly sampling a fraction from every class in the training data. MNIST & CIFAR are lower dimensional and much faster to train. Our audio dataset is more high-dimensional (sampling frequency 8-16KHz, 1s) while our EEG dataset is multivariate, higher dimensional (19 channels, 200Hz, 12s-60s) and large. We believe that in case of a large dataset, a small coreset will be sufficient, making it computationally less expensive, e.g., for our EEG experiments, we use only 2% of the training data as our coreset. We also plan to add a section to the Appendix with an ablation on coreset sizes for other datasets as well, but we hope that our success with just 2% of the large EEG training set will alleviate the reviewer’s concern.
>
> _(**Updates**: We have carried out the experiments and have added our results and description to Appendix Section E (page 18). We are happy to report that the performance is similarly high even when using only 2% or 10% of the training data and believe that it’s another strength of our method that can now be highlighted.)_

---

> > ### Author Response · Authors · 2021-11-19
> > **Complexity concerns addressed, ablations comments explained (2/2)**
> >
> > **_5. I also think the authors should extent the ablation such that they report results i) based on the spatial scores only (only distance, only similarity, both) and/or ii) the temporal score (computed directly on the embedding h) to better demonstrate the importance of each score used_**
> >
> > We thank the reviewer for the suggestion and we will plan to incorporate this as a separate ablation.
> >
> > _(**Updates**: We have carried out the experiments and have added Appendix Section F (pages 17, 18, 20, 21, 22) containing Table 8 and Figures 3, 4, 5. The added table and figures highlight the contribution of each component that allows the reader to understand where the gains are coming from. We believe that this now gives additional insight about TIME-LAPSE’s internal workings and allows a degree of explainability to our model scores.)_
> >
> > We believe we have a big part of the ablation already in our results.
> >
> > (i) only distance: One of our baseline methods (vanilla Mahalanobis) is essentially just our distance score. It is easy to see how it compares to TIME-LAPSE from Tables 1-3. It has the same trend as TIME-LAPSE in being sensitive to semantic overlap but is not high performing enough.
> >
> > (ii) (only spatial) vs (spatial+temporal): We have added new figures (Figs.3,4,5) which give a qualitative ablation between only spatial and spatial + temporal scores of TIME-LAPSE. We see from the plots that the margin between uncertainty scores of InD and OOD data is low for any of the baselines and the purely spatial TIME-LAPSE scores. Addition of the temporal scores greatly increases the separability between them.
> >
> > (iii) only temporal (directly on embeddings): We have evidence from literature to show that that hypothesis based tests such as our temporal approach do not scale well to the multivariate case [1]
> >
> > [1] Ramdas et al. On the Decreasing Power of Kernel and Distance Based Nonparametric Hypothesis Tests
> > in High Dimensions. AAAI 2015
> >
> > **_6. Results on large-scale datasets (imagenet) could strengthen the impact of the paper_**
> >
> > Thank you for the suggestion! We do believe the diversity of domains, tasks and extensive comparisons to strong baselines in our paper is its strength and shows its robustness. We will add large-scale datasets in our future work.
> >
> > We hope that we have resolved the reviewer’s concerns and hope that they can update their scores after going through our revisions. If there is any other feedback or more questions, we’d be glad to hear it.

---

### Official Review · Reviewer_54Yv · 2021-11-07

**Correctness:** 2
**Technical Novelty And Significance:** 3
**Empirical Novelty And Significance:** 2
**Recommendation:** 5
**Confidence:** 3

**Main Review:**

The strengths of this paper are two-fold. First, they have a fairly intuitive setup with a number of very desirable features (never before seen classes, high dimensional input data). Second, they have strong performance on key tasks, using a useful measure (AUROC) on a diverse set of benchmark datasets compared to a wide variety of other recent methods.

There were, however, some significant weakness in this paper.

First, is I was confused about what problem this was trying to solve - OOD, yes,  but I couldn't really tell if this was focused on detecting concept drift, data drift, anomaly detection, or something else (I think they are focused on data drift, but this is only mentioned in passing when discussing evaluation at the bottom of page 6). I believe there were technical issues in their problem setup: There should be a distinction drawn between ABSTAIN and and FLAG, which is not present in their equation. More bothersome, the temporal aspect of the problem setup was unclear. By my read of the problem setup, $x_t$ is used only as a data point, and neither the output, nor the score function have any dependence on time. The place the temporal information was encoded was in the network structure. That only makes sense if you have certain strong assumptions on your input data (e.g. regular spacing between time samples). That assumption was not stated and not checked anywhere in the paper.

Second, I have concerns about the experimental setup. The precise problem they are addressing impacts the experimental setup. My previous comment on this spacing between time points applies here. Figure 1 makes me concerned that their temporal scorer is subject to data leakage during evaluation because it is being updated based on test sets. More concerning is the setup, they are inserting change points, so in spite of the excellent data chosen, they are fundamentally using a synthetic dataset for evaluation. Because the details of that synthesis don't necessarily match the intended setup of the baseline measurements, it shouldn't be a surprise that their methods perform better. When you pick evaluation, baseline, and data, it is easy to show good performance.


**Summary Of The Paper:**

This paper was fundamentally about out of distribution detection. The authors tried to use the changing confidence around prediction values to predict changing distributions. They did so specifically for high dimensional, unstructured data, where distributions are often difficult to understand or quantify and compared their results to a number of current methods in the area on a wide variety of high dimensional data.

**Summary Of The Review:**

I think this paper has a lot of potential, but but has technical issues that should be addressed before acceptance. The experimental setup and problem statement are the primary concerns for me. What they measure seems strong, but it is the connection between those measurements and the stated goals of the paper that are the limiting factor.

---

> ### Author Response · Authors · 2021-11-19
> **Concerns addressed (Problem set up) (1/2)**
>
> We thank the reviewer for their valuable feedback & comments and for highlighting the strengths of the paper (desirable features, strong performance, diversity of datasets, tasks) and seeing the potential in it. We give responses to individual points below
>
> **_1. I was confused about what problem this was trying to solve - OOD, yes, but I couldn't really tell if this was focused on detecting concept drift, data drift, anomaly detection, or something else (I think they are focused on data drift, but this is only mentioned in passing when discussing evaluation at the bottom of page 6)_**
>
> Thank you for the feedback here. We apologise for not making it clearer before. The main aim of the paper is uncertainty estimation and we use the tasks of OOD detection and dataset shift detection as our means to validate our uncertainty estimates. To make this explicit in our paper, we have chosen a more informative title “TIME-LAPSE: Uncertainty scoring via Latent-Space Embeddings over Time”. We have also revised our introduction (section 1) to reflect that our goal is uncertainty estimation while OOD & drift detection are the concrete tasks to help us justify the quality of our uncertainty  estimates. We hope this resolves the confusion. Please let us know if you have any remaining concerns.
>
> **_2. I believe there were technical issues in their problem setup: There should be a distinction drawn between ABSTAIN and and FLAG, which is not present in their equation._**
>
> Thank you for the comment. However, we wish to clarify that the equation in Section 3.1 is during **inference** and not training. We have updated the text in Section 3.1 to explicitly mention that this is during inference. Thus, there is no requirement for a distinction to be made between ABSTAIN and FLAG. Rather, they are choices available to the user when they obtain the predictive uncertainty scores $s(x_t)$ from TIME-LAPSE during inference. This choice does not affect the problem or the way the uncertainty score is generated. In fact, their choice depends on the downstream task and application. For e.g., in OOD detection, when the uncertainty score is high, “FLAG as OOD” makes more sense. In a clinical workflow, if the uncertainty score is high, ABSTAINing from prediction and alerting the clinicians instead makes more sense. We hope that this clarifies the confusion
>
> **_3. The temporal aspect of the problem setup was unclear. By my read of the problem setup, it is used only as a data point, and neither the output, nor the score function have any dependence on time. The place the temporal information was encoded was in the network structure. That only makes sense if you have certain strong assumptions on your input data (e.g. regular spacing between time samples). That assumption was not stated and not checked anywhere in the paper._**
>
> Thank you for this feedback. However, we believe there is misunderstanding about our work here.
>
> a) We want to clarify that our network structures (trained encoders) do not have any temporal information pertaining to TIME-LAPSE encoded in them. They treat a datapoint (an image, an audio clip, an EEG clip) individually and output a latent-space embedding. Whether the datapoint itself has any temporal information in it (audio/EEG clip) or not (image), has no bearing on TIME-LAPSE.
>
> b) Our spatial score function takes the embeddings provided by the encoder during inference individually and produces their scalar spatial uncertainty scores (explained in Fig. 1, Section 3.2.1). Our temporal score function is the place where temporal information is encoded. It takes in a small sequence of spatial scores previous to the current sample and produces the scalar temporal uncertainty score (explained in Fig. 1, Section 3.2.2). We make no assumptions about input data such as having regular spacing between time samples.
>
> c) We’d like to clarify that we’re using the terms “spatial” and “temporal” in a different context compared to the video community which could be the source of the confusion here. Spatial in TIME-LAPSE refers to the **latent space** and not the space of the datapoint itself (as in a video where space refers to the dimensions within a video frame). Similarly, temporal in TIME-LAPSE is not time within the data point (as in a video or a time-series clip) but across **different data points**. We believe the term “spatial-temporal” could be the origin for this misunderstanding and apologise for using overloaded terminology. We have updated our paper to remove that term to avoid confusion. Please let us know if there are any lingering questions / concerns regarding this point after reading our updated intro.
>
> We hope that we have resolved the reviewer’s concerns about the problem set up and hope that they can update their scores after going through our revisions. If there is any other feedback or more questions, we’d be glad to hear it.

---

> > ### Author Response · Authors · 2021-11-19
> > **Concerns addressed (Experimental set up) (2/2)**
> >
> > **_4. I have concerns about the experimental setup. The precise problem they are addressing impacts the experimental setup. My previous comment on this spacing between time points applies here. Figure 1 makes me concerned that their temporal scorer is subject to data leakage during evaluation because it is being updated based on test sets. More concerning is the setup, they are inserting change points, so in spite of the excellent data chosen, they are fundamentally using a synthetic dataset for evaluation. Because the details of that synthesis don't necessarily match the intended setup of the baseline measurements, it shouldn't be a surprise that their methods perform better. When you pick evaluation, baseline, and data, it is easy to show good performance._**
> >
> > Thank you for the feedback. We apologise for not making the experimental set up clearer in the first place. We have updated Sections 4.1, 5.3 and Appendix B.3 to make the clarifications more explicit.
> >
> > We wish to alleviate concerns raised by the reviewer and emphasize that
> >
> > (a) We do not have a problem of data leakage in our temporal scoring function. Test data used for evaluation  have never been seen by the encoder, spatial scorer or temporal scorer. Datapoints from the test sets are presented to our framework during inference. We do not update the temporal scores of previously tested data using current temporal scores, nor do we make use of temporal scores from previous inputs at any point. We hope that this resolves the data leakage concern on the part of the reviewer. We believe that the updated clarifications to the experimental set up (written below) will automatically remove this worry. Please let us know if you have any further questions.
> >
> > (b) We use the change point set-up for our drift detection downstream task and not for our OOD task. We have updated the title of Section 5.3 and the descriptions in Section 4.1 and 5.3 to reflect this clarification. For OOD detection, we just use unseen test sets for evaluation. For drift detection, since datasets (by definition) are finite, we need to sample from and introduce distributional shifts synthetically to emulate long data streams with real-life drift conditions. We point out that this is common practice in the field [1,2]. Furthermore, the synthetic streams are generated in a principled manner as explained in Section 4.1, 5.3 by randomly selecting probability of distribution change & change point intervals. We perform N = 1000 trials to ensure robustness. We also refer to newly added figures Appendix Figs. 3,4,5 which can illustrate how TIME-LAPSE performs compared to other methods. We hope this clarifies the reviewer’s question. Please let us know if you have any further questions.
> >
> > [1] Gama et al. A survey on concept drift adaptation. ACM Computing Surveys (2014)
> > [2] Liu et al. Change-point detection in time-Series data by relative density-ratio estimation. Neural Networks (2013)
> >
> > (c) We compare all baselines and our methods across all datasets and tasks in the same manner. We believe that TIME-LAPSE’s strength shows up via its strong performance across all tasks and domains. We explicitly chose a diverse range of tasks (easy: MNIST, hard: clinical EEGs), domains (audio, vision, high dimensional medical time-series) and 6 strong, widely-accepted baselines (MSP, predictive entropy, KL divergence with uniform distribution, ODIN, vanilla Mahalanobis, Test-time Dropout) to test and benchmark our methods as well as all baselines rigorously.
> >
> > We hope that we have resolved the reviewer’s concerns about the experimental set up and hope that they can update their scores after going through our revisions. If there is any other feedback or more questions, we’d be glad to hear it.

---

### Author Response · Authors · 2021-11-21
**Overall response to AC and all reviewers (1/2)**

Dear AC and Reviewers,

We would like to thank all the reviewers for their close read of our paper, their insightful comments and the constructive feedback. We were encouraged by the enthusiasm for our concept and idea as well as the strengths highlighted by the reviewers (desirable features, strong performance, diversity of datasets, tasks: reviewer 54Yv; promising direction, semantics preserving and state-of-the-art results: reviewer o7sa; interesting ideas, use of temporality, diversity of datasets, tasks: reviewer CrvH; relevance of the problem, thorough experimentation, well-organized & well-written: reviewer qgxp).

Many important suggestions were made and thoughtful questions were posed which ultimately strengthened our paper significantly. We have considered each comment carefully and have revised our paper accordingly. Please find our detailed replies to individual reviewers below in our responses.

We are happy to highlight the major changes in our manuscript here that resulted from the review process and improved the quality of our work tremendously:

**Experiments**

**Means & Standard deviations for tables**: We thank reviewer qgxp for bringing this up. We have updated the manuscript to report means and standard deviations across 5 random runs to all our experiments and have verified that our results and conclusions remain the same. We have updated Tables 1, 2, 3,  4, 5, 6, 7 (pages 7, 8, 14). We report only means in the main manuscript due to our space constraint. But we report the expanded tables with means and standard deviations in our Appendix (Tables 4,5,6; pages 14,15).

**Ablations: Coreset Size**: We thank reviewers o7sa and CrvH for their suggestion on having a coreset size ablation to alleviate concerns about the trade-off between computational complexity and performance. We have carried out the experiments and have added our results and description to Appendix Section E (page 18). We are happy to report that the performance is similarly high even when using only 2% or 10% of the training data and believe that it’s another strength of our method that can now be highlighted.

**Ablations: Contributions of individual scores in TIME-LAPSE** (distance vs similarity, spatial vs temporal, only temporal): We thank reviewer o7sa for their suggestion on having an ablation study to understand the effect and contribution of each component in TIME-LAPSE. We have carried out those experiments and have added Appendix Section F (pages 17, 18, 20, 21, 22) containing Table 8 and Figures 3, 4, 5. The added table and figures highlight the contribution of each component that allows the reader to understand where the gains are coming from. We believe that this now gives additional insight about TIME-LAPSE’s internal workings and allows a degree of explainability to our model scores.

**Temporal Experiments Description, Clarifications and Explanations**: We thank reviewers 54Yv, o7sa, CrvH for giving us valuable feedback that our temporal experimental setup could be expanded and be made more clearly and thoroughly. We have updated the descriptions of Section 4.1 (page 6), 5.3 (page 8) and Appendix B.3 (page 16, 17) to be more clear with additional details on hyperparameters and design choices that should clarify all the questions and confusion the reviewers had at first.

We describe further changes in part 2/2 below.

---

> ### Author Response · Authors · 2021-11-21
> **Official response to AC and all reviewers (2/2)**
>
> (official response continuation)
>
> **Problem Description / Introduction / Terminology**
>
> We thank all the reviewers for their questions regarding our problem description, introduction and terminology. We believe that our updates to the manuscript will now make it easier for readers to understand the nuances raised by the reviewers.
>
> We have updated our title and introduction to clarify that the main aim of the paper is **uncertainty estimation**. We use the tasks of OOD detection and dataset shift detection as our means to validate our uncertainty estimates. To make this explicit in our paper, we have chosen a more informative title “TIME-LAPSE: Uncertainty scoring via Latent-Space Embeddings over Time”. We have also revised our introduction (Section 1) to reflect that our goal is uncertainty estimation while OOD & drift detection are the concrete tasks to help us justify the quality of our uncertainty  estimates.
>
> We realised the reviewers had common comments regarding terminology that could potentially confuse the reader such as the term “spatial-temporal”, which is largely overloaded with video domain semantics. We have updated the manuscript to make it explicit that we are using the terms “spatial” and “temporal” in a different context here. Spatial in TIME-LAPSE refers to the **latent space** and not the space of the datapoint itself (as in the case of a video where space might refer to the dimensions within a video frame). Similarly, the temporal dimension in TIME-LAPSE is not time within the data point (as in a video or a time-series clip) but across **different data points**. We have thus removed any explicit mention of the term “spatio-temporal” throughout the manuscript and instead use “latent-space embeddings over time” to remove any confusion. We believe that while one of our strengths (reviewer qgxp) is strong writing and organization, the updates to our paper make it even more so.
>
> **General**: We have added Ethics and Reproducibility statements to our paper.
>
> We believe that TIME-LAPSE addresses an important problem, shows significant gains over baselines across a diverse set of benchmarks & tasks and would be crucial for developing robust models in safety-critical domains. We would again like to thank all reviewers for their time and feedback, and we hope that our changes adequately address all concerns.

---

### Decision · Program_Chairs · 2022-01-20

**Decision:**

Reject

**Comment:**

This paper has been reviewed by four experts. Their independent evaluations were all below the acceptance threshold citing various issues ranging from disconnection between stated goals of the presented work and the means in which the approach was evaluated, to doubts about the scalability of the proposed approach, to the lack of clarity regarding the actual novelty of the approach given some key missed references, to name a few items of criticism. Most reviewers were impressed with the empirical performance achieved in the conducted experiments, and one of the reviewers raised their mark in response to the author's rebuttal. Yet, the overall evaluation places this work as it stands now below the threshold for ICLR acceptance. I would like to encourage the authors to continue pushing their promising endeavor and systematically incorporating the feedback received here to improve the overall quality of this work.